# Geospatial Intelligence and Machine Learning Technique for Urban Mapping in Coastal Regions of South Aegean Volcanic Arc Islands

Pavlos Krassakis [1,2,*], Andreas Karavias [2], Paraskevi Nomikou [3], Konstantinos Karantzalos [4], Nikolaos Koukouzas [2], Stavroula Kazana [3] and Issaak Parcharidis [1]

1   Department of Geography, Harokopio University of Athens, El. Venizelou 70, 17671 Athens, Greece
2   Centre for Research & Technology Hellas (CERTH), 15125 Athens, Greece
3   Department of Geology and Geoenvironment, National and Kapodistrian University of Athens, Panepistimioupoli Zografou, 15784 Athens, Greece
4   Department of Topography, School of Rural and Surveying Engineering, Zografou Campus, National Technical University of Athens, 9 Heroon Polytechniou Str., 15780 Athens, Greece
*   Correspondence: krassakis@certh.gr or pkrassakis@hua.gr

**Abstract:** Coastal environments are globally recognized for their spectacular morphological characteristics as well as economic opportunities, such as fisheries and tourism industries. However, climate change, growth in tourism, and constant coastal urban sprawl in some places result in ever-increasing risk in the islands of the South Aegean Volcanic Arc (SAVA), necessitating thoughtful planning and decision making. GEOspatial INTelligence (GEOINT) can play a crucial role in the depiction and analysis of the natural and human surroundings, offering valuable information regarding the identification of vulnerable areas and the forecasting of urbanization rates. This work focuses on the delineation of the coastal zone boundaries, semi-automatization of Satellite-Derived Bathymetry (SDB), and urban mapping using a machine learning algorithm. The developed methodology has been implemented on the islands of Thira (Santorini island complex) and Milos. This study attempts to identify inaccuracies in existing open-source datasets, such as the European Settlement Map (ESM), as a result of the unique combination of the architectural style and bare-soil characteristics of the study areas. During the period 2016–2021, the average accuracy of the developed methodology for urban mapping in terms of the kappa index was 80.15% on Thira and 88.35% on Milos. The results showed that the average urbanization expansion on specified settlements was greater than 22% for both case studies. Ultimately, the findings of this study could contribute to the effective and holistic management of similar coastal regions in the context of climate change adaptation, mitigation strategies, and multi-hazard assessment.

**Keywords:** GEOINT; SAVA; random forest; coastal zone; machine learning

## 1. Introduction

According to the EU blue economy report 2021, coastal areas have a crucial economic role in the Greek economy, with the financial sector accounting for 14.2% of jobs and 5.2% of Gross Domestic Product (GDP). Specifically, coastal tourism is the greatest contributor, accounting for 13% of Gross Value Added (GVA) and 3.8% of employment [1]. More than one-third of Greece's population lives within two kilometers of a coastline, and more than 85 percent of the country's industrial activity, which is mainly tourism, is located in coastal areas. Additionally, coastal and deltaic plains serve as fertile land for agricultural endeavors [2]. The development of coastal zones and the implementation of management strategies in these regions are important not only for Greece but also for the other Mediterranean countries. Furthermore, the economic influence that coastal regions have on the growth of Greece makes them highly valuable. Compared to the 20.6 million

tourists that visited the country in 2010, Greece welcomed over 32 million visitors in 2019 [3]. According to the United Nation World Tourism Organization (UNWTO), Greece is the eighth biggest tourist destination in the world in terms of the largest surplus in the travel balance [4].

Therefore, under the prism of the natural environment, Greece is also a unique physical laboratory since it is one of the most seismically and volcanically active areas in Europe. The country's geological history is associated with the Hellenic Subducting System (HSS) and the regional tectonics that have produced several volcanic edifices, with the most intriguing geomorphological agents being the Methana, Milos, Santorini, and Nisyros active volcanic centers located in the South Aegean Sea.

Considering the climate change in Mediterranean countries, the loss of coastal land, which is an area where many human activities are concentrated, is a considerable consequence and is strongly related to Sea Level Rise (SLR) and coastal erosion. Taking into account SLR projections for the 21st century in Mediterranean regions, the wetlands and dry lands are predicted to face the greatest negative impacts [5]. According to a recent study by Vousdoukas et al. [6], extreme sea levels (ESLs) in the European region could rise by 1 m or more by the end of the century, exacerbating the impact of coastal hazards. In addition, recent research estimates that the Mediterranean Sea's projected SLR by 2050 will be up to 25.6 cm [7,8]. Coastal erosion is generally caused by a combination of physical and anthropogenic forces operating on various scales. Winds and storms, nearshore currents, SLR, and the slope angles of the coastal zone are the most important natural causes [2,8]. On the other hand, human factors that cause coastal erosion are related to coastal engineering and land uses such as extended urbanization for industrial and tourism activities.

Taking into consideration the aforementioned issues, the monitoring of coastal regions is a very important process due to the fact that its results may contribute substantially to decision-making processes related to coastal urban planning and the minimization of natural hazard impacts and losses. This can be accomplished using the GEOspatial INTelligence (GEOINT) approach, which is the analysis and combination of satellite imagery and geospatial information in an effort to describe, interpret, and anticipate the physical features, human impact, and activities within a spatiotemporal environment [9,10]. The GEOINT definition originates from the United States' law and army services and describes the ability to create and present geospatial knowledge by collecting, identifying, and manipulating data for the decision-making environment [10]. GEOINT applications could be used to monitor, conceptualize, and predict the exposure of coastal urbanization to natural hazards related to SLR, floods, tsunamis, or coastal erosion by analyzing and interpreting geospatial and Earth Observation (EO) data. GEOINT applications can be applied in order to analyze the coastal changes and predict erosion trends. Imagery information (EO data) combined with geospatial information can be used to monitor, conceptualize, and predict the exposure of coastal urbanization to natural hazards related to SLR, floods, or even tsunamis. However, the global or continental-scale open geospatial datasets usually face generalizations, failure, or are not updated frequently. Examples of open-source datasets are the European Settlement Map (ESM), based on satellite images using Machine Learning (ML) techniques, and OpenStreetMap (OSM), which is a world-class public geodatabase. This paper illustrates this standardized approach of ESM products, which is facing difficulties in the extraction of urban environments in the South Aegean islands. Spectral similarities across classes or spectral values influenced by topographical features such as slope, height, or sun radiation are the most common causes of misclassification [11,12]. To address this issue, this manuscript aims to create an innovative approach leveraging a machine learning algorithm in the ArcGIS environment in order to extract with high accuracy the urban environment of the studied islands in an effort to reduce misclassification in open-access datasets.

During the last decades, researchers have developed a number of approaches for extracting information about the urban built-up areas from EO images using vegetation indices, temperature inversion, and impervious surface density [13–15]. In recent years,

different approaches using machine learning (ML) have been implemented in various fields of EO and natural hazard monitoring [16–22]. The application of machine learning has led to the production of very precise results, especially in the identification of urban areas and spatial patterns [23,24]. In addition, there are several Maximum Likehood methods and ML algorithms such as support vector machines (SVMs) and Random Forest (RF) that have contributed to the improvement in the research progress concerning the mapping of urban built-up areas [25,26]. Previous research has proven that, contrary to parametric classifiers and other models, RF is utilized by higher classification accuracy on high-resolution dimensional data [27].

As stated in the first part of the introduction, coastal areas around the world are identified as the most valuable zones for human activities. These transition areas have also been marked as physically dynamic buffer zones between land and sea [28]. Coastal zones have been characterized as vulnerable areas to potential hazards and climate change by coastal hazard specialists [29]. According to other publications and technical reports from national organizations, the boundaries of coastal areas are frequently delineated in regional scale using specific criteria and linear distances from the coastline. A good example is the report of the National Oceanic and Atmospheric Administration (NOAA) that sets the coastal zones buffers for (i) California at up to 914.4 m inland from the mean high tide, (ii) New York at 152.4 m inland from the coastline, and for (iii) Puerto Rico at 1000 m [30]. In addition, the Integrated Coastal Zone Management in the Mediterranean (ICZM) describes the coastal zone as a dynamic geomorphologic area characterized by the diverse interactions between the marine and land parts. This interactivity occurs in the form of complex natural and manmade systems involving biotic and abiotic components such as local socioeconomic activities. Moreover, Article 8 of the ICZM protocol specifies a width zone of 100 m [31] along the highest winter waterline in order to set the limits of the coastal region in the Mediterranean zone.

In particular, this research tries to adopt relevant literature and Article 6 of the National Report of Greece on Coastal Zone Management in the context of the Recommendation on Integrated Coastal Zone Management [32], where the critical boundaries of the seaward part of the coastal zone of the Aegean region are established at the depth value of 10 m. On the other hand, for the determination of the landward bound, the Extended Low Elevation Coastal Zone (E-LECZ) [17] has been analyzed and implemented in order to set its boundary up to the elevation contour of 20 m.

Under this frame, this work attempts to combine the aforementioned definitions, technical reports, and literature in an attempt to specify and visualize the spatial boundaries of the coastal zone in Thira and Milos islands. Therefore, the findings could play a key role in the future as a baseline for multi-hazard approaches and early warning systems in the wider area of the determined coastal zones. Consequently, this study tries to monitor the spatiotemporal evolution of the urban fabric between 2016 and 2021, using open-source data in order to highlight areas with the highest urbanization rate within the boundaries of the specified costal units. The objectives of this work are divided into a threefold process: (i) the determination of the coastal boundaries in Milos and Thira, (ii) the semi-automation of the GEBCO-like empirical approach for Satellite Derived Bathymetry, and (iii) the development of an innovative semi-automatic geoprocessing workflow for the accurate mapping of the urban fabric on the unique environment of the southern Aegean islands using a machine learning technique in ArcGIS software.

## 2. Materials and Methods

### 2.1. Study Areas

The islands of Thira (Santorini Island complex) and Milos, both of which are administratively part of the South Aegean Region (which extends from 24°19′ E, 36°42′ N to 25°29′ E, 36°22′ N), are located inside the margins of the islands of the South Aegean Volcanic Arc (SAVA).

The SAVA is positioned approximately 150 km north of the Hellenic sedimentary arc in the South Aegean Sea and is characterized by both subaerial and submerged volcanism. Its margins extend from Crommyonia and the centers of the Saronic Gulf in the west to the volcanic field of Kos–Yali–Nisyros in the east [33,34] (Figure 1). The SAVA has been developed during the last 5 Ma, in the continental crust of the Hellenic Subduction System (HSS), due to the northward subduction of the last remnant of the oceanic crust of the African plate, beneath the southern edge of the active margin of the European plate [35,36].

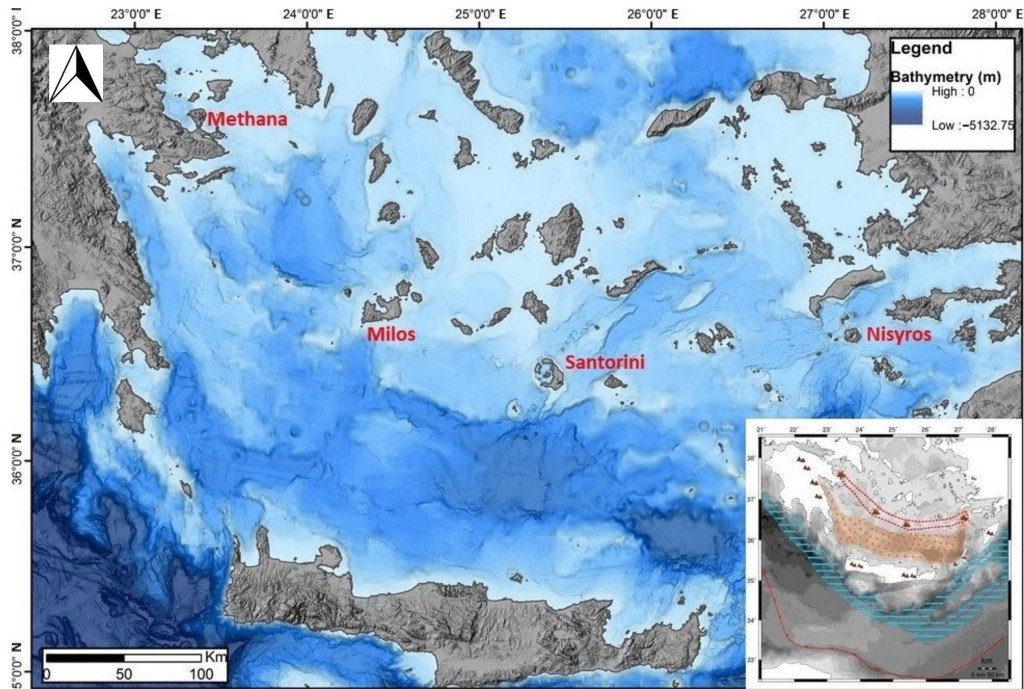

**Figure 1.** Synthetic Topographic Map of the South Aegean Sea depicting the islands of the SAVA. Inset map: Map of the HSS where the East Mediterranean Chain (bottom grey area), the Hellenic trench (blue lines), the Island arc (red triangles), the Cretan Back-arc Basin (orange dots), and the SAVA (top red triangles) are indicated (modified from [34]).

2.1.1. Milos

The island of Milos is the most southwestern of all the Cyclades Island complex, and it is the largest island in the Milos volcanic complex, which includes the islands of Kimolos, Polyegos, and Antimilos. The general geomorphology of Milos Island is characterized by a predominantly steep and uneven coastline, as well as an inner large central bay (Gulf of Milos), which offers a natural harbor and gives the island the appearance of a horseshoe shape (Figure 2). Specifically, the elevation range of the island is typically small, consisting of low hills and mountains; the eastern part of the island has a more regular terrain than the western part of the island. Profitis Ilias summit is the highest in Milos (751 m) [37]. From the Pliocene up until historical times (200 BC–200 AD), the island of Milos has a long history of volcanic activity for over 3 Ma inside the SAVA [38,39]. In the context of coastal analysis, Milos has a coastline with a total length of 147 km. The coastal area of the island has a total extent of 122.36 km$^2$, including the offshore section (23.36 km$^2$) up to the depth of 10 m, as well as the onshore area (99 km$^2$) up to an elevation of 20 m (see Section 2.3.1).

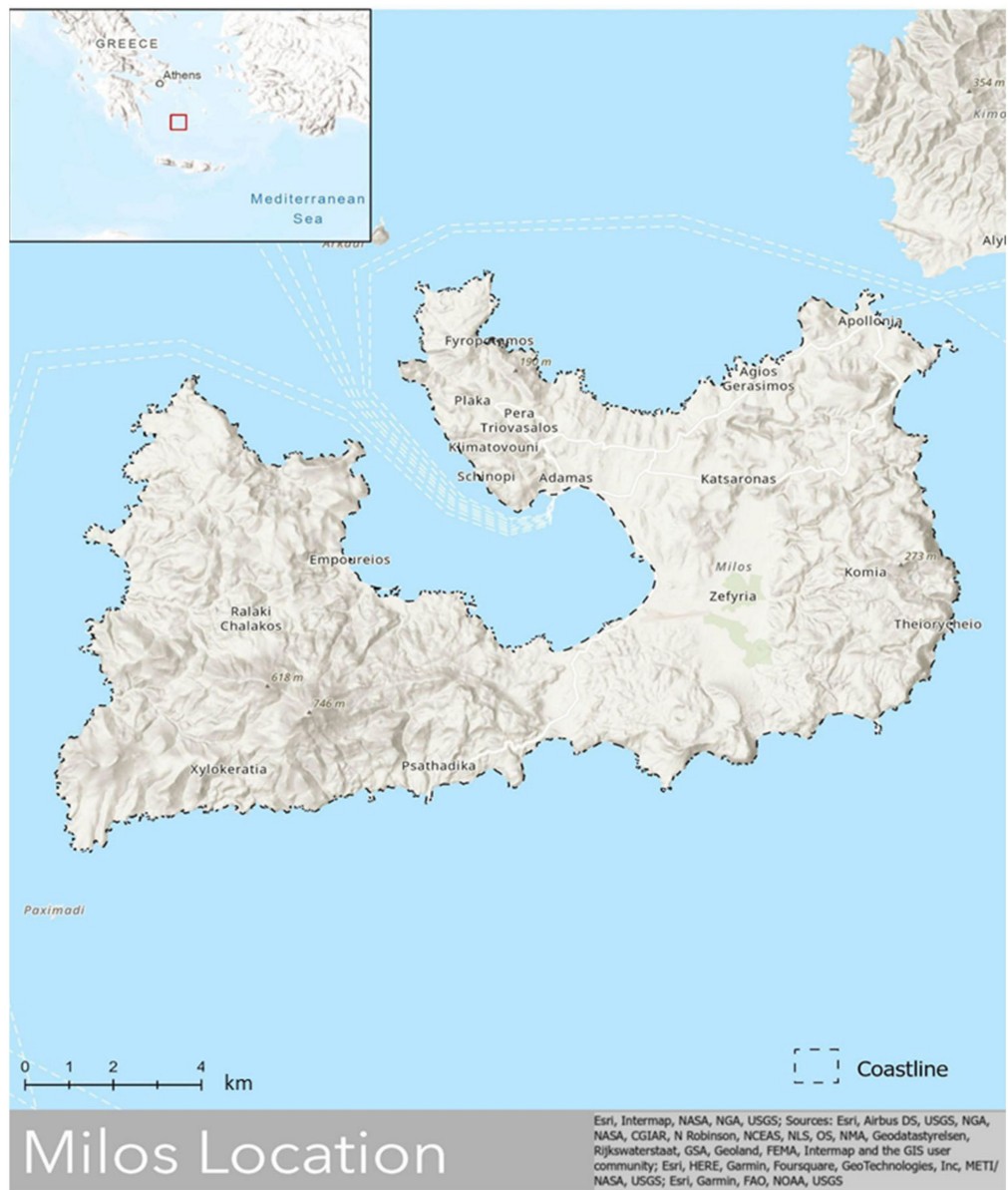

**Figure 2.** Topographic map of Milos Island. Inset map: Location of Milos in relation to the Hellenic region (ESRI basemap).

### 2.1.2. Thira (Santorini Island Complex)

The Santorini Island complex is located in the center of the Cyclades group of islands and is part of the Christiana–Santorini–Kolumbo volcanic group. Thira is the largest of the five islands of Santorini Island complex (Figure 3). The islands are comprised of rugged terrain (such as calderas and craters), while their outside is composed of smooth terrain (erosional features and sculpted ignimbrites). The volcanic topography generated by effusive and explosive landforms, including cones, domes, and craters, may be found on the main island, as well as the Kolumbo volcanic chain and Christiana islands.

Specifically, the elevation range of the island is typically small, consisting of low hills and mountains; the eastern section of the island has a more regular terrain than the western part of the island [39]. In terms of coastal geospatial analysis, Thira has a coastline with a total length of 147 km. Its coastal area has a total extent of 86.48 km$^2$, including the offshore section (14.48 km$^2$) up to the depth of 10 m, as well as the onshore region (72 km$^2$) up to an elevation of 20 m (see Section 2.3.1).

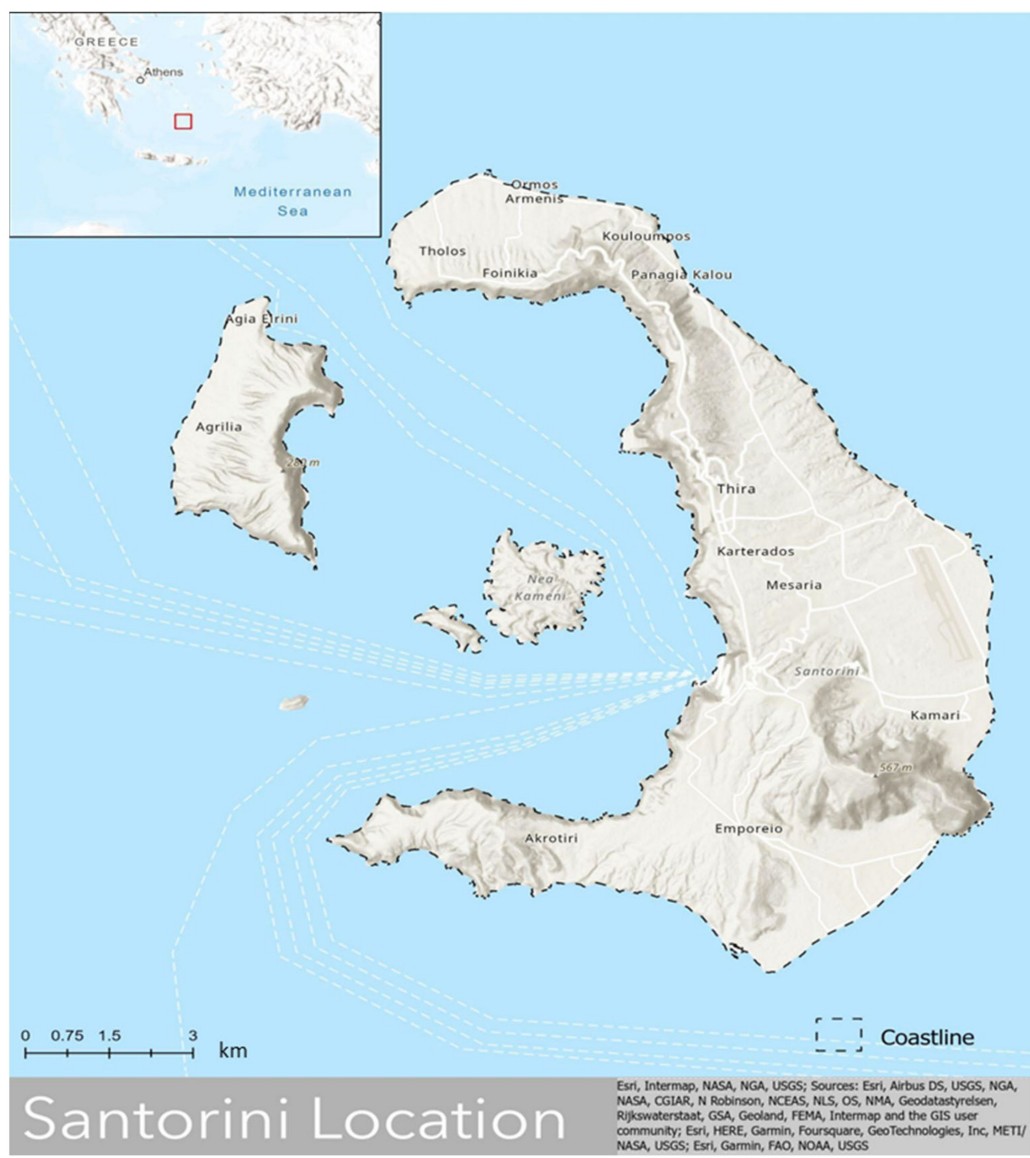

**Figure 3.** Topographic map of Santorini Island complex. Thira is the largest island on the right part of the image. Inset map: Location of Santorini Island complex regarding Hellenic region (ESRI basemap).

*2.2. Data*

The dataset processing of this study is based on open-access products of the Copernicus programme and other open-source databases that are described in the following section. Specifically, the final products were produced by utilizing free satellite images of Sentinel-2 mission and datasets from Corine Land Cover (CLC) 2018 [40], European Settlement Map (ESM) 2015 [41], a Digital Elevation Model (DEM) from Shuttle Radar Topography Mission (SRTM) [42], and OpenStreetMap (OSM) [43] in an effort to verify and validate the findings of our study. The different varieties of datasets as well as their technical specifications that have been incorporated in this study are outlined in more detail in Table 1.

**Table 1.** Dataset used for Urban Mapping in Coastal Regions of Hellenic Volcanic Arc Islands.

| Data Usage | Data Sets | Data Source | Spatial Scale | Temporal Scale | Primary Format |
|---|---|---|---|---|---|
| Coastline, Bathymetry and Urban environment extraction | Sentinel-2 | Copernicus | 10 m | 2016–2021 | Raster (grid) |
| Elevation and coastal areas determination | SRTM-1sec (DEM) | USGS | 30 m | - | Raster (grid) |
| Road network | Road Grid | Open Street Maps | - | - | Vector (polylines) |
| Land Uses | Corine Land Cover | Copernicus | 100 m | 2018 | Vector (polygons) |
| Urban environment validation | ESM | Copernicus | 10 m | 2014–2016 | Raster (grid) |

In order to produce the isobath of 10 m and the coastal spatiotemporal urbanization, two optical images of Sentinel-2 mission were obtained for every island on the time period from September 2016 to September 2021. The Sentinel-2 mission consists of two polar-orbiting satellites (A and B) phased at 180 degrees apart in the same sun-synchronous orbit. The latter were launched on 23 June 2015 and 7 March 2017, respectively, having a wide swath width (290 km) and high revisit time of 5 days with two satellites in cloud-free conditions.

Corine Land Cover (CLC) is a database of Copernicus Land Monitoring Service (CLMS) concerning land cover and land use in European countries that aids in the development of environmental policy. The CLC is divided into 44 classes based on five main land cover types (artificial surfaces, agriculture, forests and seminatural regions, wetlands, and water), each with a geometric detail of 25-hectare minimum mapping unit and 100 m minimum mapping width [33]. The European Settlement Map (ESM) is produced in the framework of the Global Human Settlement Layer (GHSL) project of the European Commission that maps human settlements in Europe made of Copernicus very high-resolution image collection from a variety of satellite sensors which range from 2014 to 2016 [44]. The datasets of the ESM have been generated using machine learning (ML) [45] classification method that has been combined with textural and morphological features to describe the human settlements.

For the purpose of extracting contour lines, the Shuttle Radar Topography Mission (SRTM) 1 arc-second DEM was implemented. SRTM 1-arcsec has a 30 m spatial resolution produced by the National Aeronautics and Space Administration (NASA) and the National Geospatial-Intelligence Agency (NGA) using Synthetic Aperture Radar (SAR) Spaceborne images [42]. Further data were openly accessed on EarthExplorer (EE) [46] platform developed by the United States Geological Survey (USGS).

OpenStreetMap (OSM) is an open access dataset that includes geospatial products related to land uses, transportation network and infrastructures in global scale [43]. The dataset is continuously updated by users around the world providing a crucial geospatial information that can be used for commercial and research purposes. In this study, the primary and the secondary classes of the road network were merged and implemented as the main road network of the Cyclades islands.

### 2.3. Methodology

The methodology of this study is separated into threefold processing: (1) the determination of the coastline boundaries introducing a schematic cross-section figure based on specific reports and literature for the South Aegean areas; (2) the implementation of the Satellite-Derived Bathymetry adopting GEBCO-like workflow in order to use Model Builder to semi-automate the geoprocessing workflow in GIS environment; and (3) the development of a semi-automated process in order to extract and identify the spatiotem-

poral evolution of the built-up area in the coastal areas of Thira and Milos islands in the 2016–2021 time period.

According to the upper and left part of the workflow (Figure 4), the first step was the pre-processing phase during which all the satellite acquisitions that were obtained in 2016 and 2021 were converted from Level-1C to Level-2A. More analytically, for the pre-processing of satellite images, the free software Sentinel Application Platform (SNAP) provided by the European Space Agency (ESA) was implemented during the initial stage. This step was important in order to obtain Bottom of Atmosphere (BOA)-corrected transforms from Top of Atmosphere (TOA) [47] reflectance of Level-1C products using the atmospheric correction algorithm Sen2Cor [48] within the environment of SNAP. In the next step, a cloud mask was applied in order to minimize the impact of water vapors and clouds on images. However, the unique Cycladic architecture of white side by side houses, the volcanic formations and the existing open-pit mines of the two study areas mirrored the similar reflectance values with the clouds (Figure 5). In an effort to avoid this issue, the cloud masking step was omitted from the presented workflow.

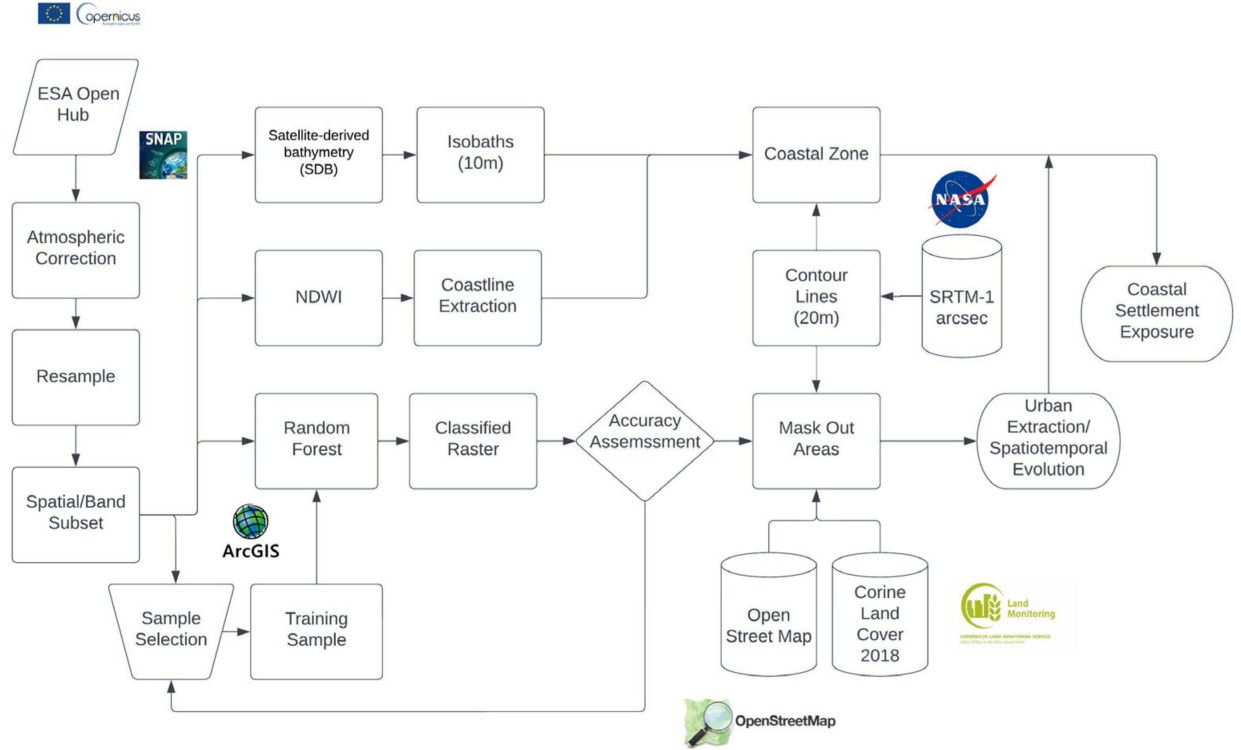

**Figure 4.** Workflow of the satellite and geospatial datasets that were implemented for the three-fold process adopting GEOINT and ML.

The following step was the resampling of multi-size bands to a single-size of 10 m and the spectral and spatial subset concerning the polygon areas on the volcanic islands. In this particular stage, only 11 bands of the visible spectrum were implemented and specifically (Blue, Green, and Red), near-infrared (NIR), and short-wave infrared (SWIR) channels (Figure 6). Regarding the presented workflow (Figure 4) and the middle-center part of it, the corrected clipped satellite images were processed for the creation of the Normalized Difference Water Index (NDWI) datasets, as means to define the coastal boundaries of Milos and Thira regions. The trace of the coastline was utilized for the geometrical calculation of the polygon area between the selected onshore region and the elevation of 20 m taking into account the Extended Low Elevation Coastal Zone. In accordance with the following and bottom part of the schematic workflow (Figure 4) the Random Forest algorithm was applied and repeated several times for the two case studies. Then, the derived results were

quantified and analyzed within the boundaries of the landward zone in order to extract and identify the urban spatiotemporal evolution between 2016 and 2021 time period.

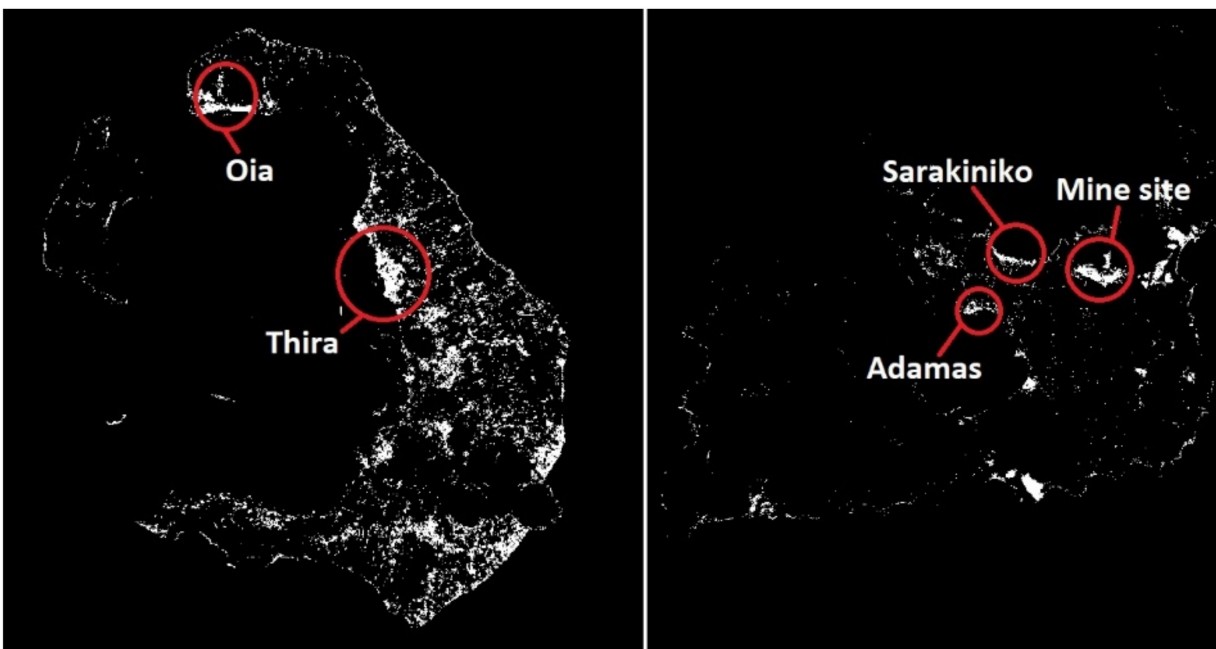

**Figure 5.** The image represents the false categorization of urban, open-pit mines and volcanic geological formations using the cloud mask in two study areas (left image: Thira; right image: Milos). The white color represents cloudy regions, while the black color depicts regions without clouds. The red circles highlight several examples of sites incorrectly classified as clouds.

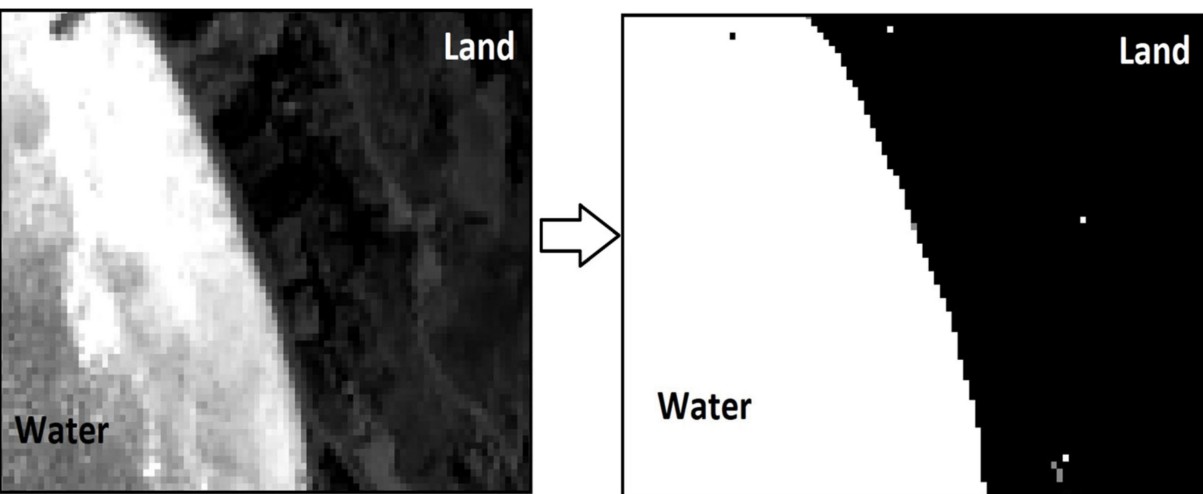

**Figure 6.** NDWI image of a processed multispectral satellite image of Milos Island. Water is represented by the bright pixels (positive values), whereas soil or land surfaces are depicted by the dark pixels (with zero or negative values).

### 2.3.1. Coastline Delineation

As mentioned before, the coastline which is the boundary line of sea and land, is vulnerable to natural processes, such as coastal erosion-accretion and sea level changes, as well as human activities. In our work, due to the regional scale of the analysis, the coastline and shoreline are identical. The delineation of the coastline is an initial and key step for sustainable management and development of a coastal region [49–51]. In addition, due

to the high and time-consuming cost of the traditional ground surveying approach, open-access remote sensing data were analyzed based on Sentinel 2 images. Specifically, for the coastline extraction, the Normalized Difference Water Index (NDWI) formula developed by McFeeters [52] was successfully implemented (Figure 6). NDWI is a satellite-derived index that utilizes the Near-Infrared (NIR) and Green wavelengths to enhance the presence of water bodies in the images. There are several studies showing that the index can be applied effectively to determine the dividing line between water and soil [49–52]. The NDWI is calculated as follows:

$$NDWI = \frac{Green - NIR}{Green + NIR} \tag{1}$$

where *Green* is Band 3 and *NIR* is Band 8 of the Sentinel-2 satellite system. The range of variability is between −1 and +1, where values of *NDWI* which are greater than 0.3 correspond to water bodies [53].

However, the threshold varieties on every image with small deviations from the reference value that has been applied were ranging from 0.1 to 0.2 (Figure 6). The NDWI threshold value implemented in the images of 2021 in order to extract the coastline in both islands.

Furthermore, and taking into account the aforementioned definitions and the National Report of Greece on Coastal Zone Management, the isobath value of 10 m (Figure 7) was established as the critical boundary of the seaward region. In accordance with the literature [54], the submerged segment was defined at 10 m due to the assumption that the waves in the Aegean Sea rarely exceed the arithmetic value of 6 m in height.

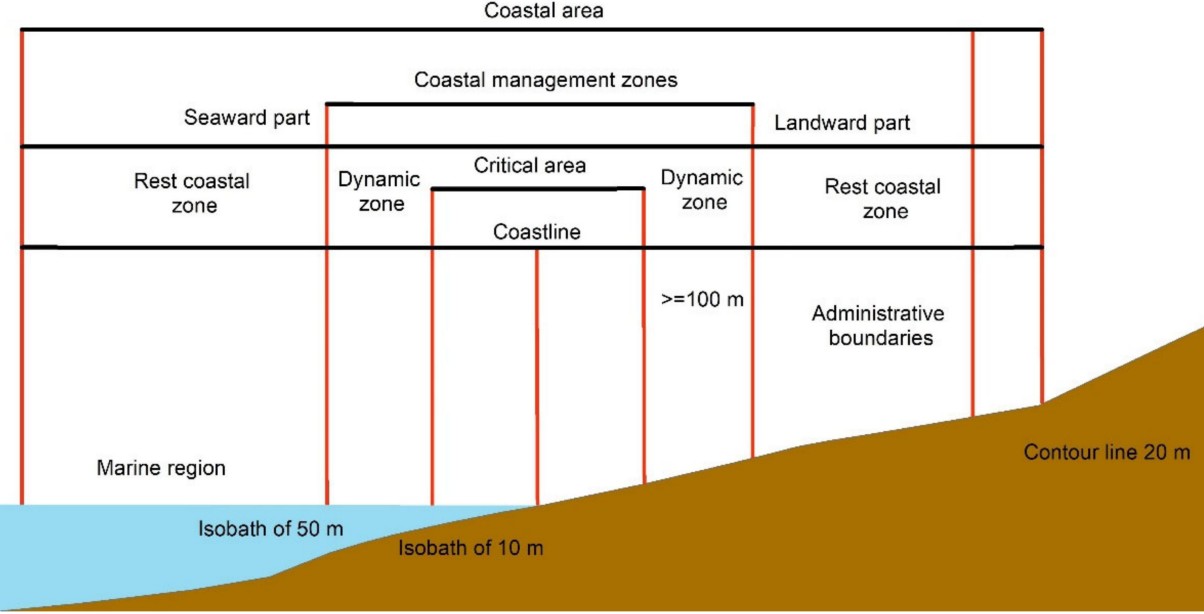

**Figure 7.** Cross-section of the proposed coastal unit boundaries integrating the developed workflow within the boundaries of the Aegean Sea.

On the other hand, and taking into consideration the Extended Low Elevation Coastal Zone (E-LECZ), the onshore limit was set at the elevation contour of 20 m [17]. According to this reference, this limit is strongly related to the assumption that this zone includes the most important urban growth into long-term adaptation and land-use planning (Figure 7). Thus, this zone in low-lying coastal regions can be considered very important for the development and implementation of future effective techniques for reducing potential coastal hazards. According with this hypothesis, in that region will take place the most significant urban growth until 2100 based on maximum urban exposure scenario [18].

### 2.3.2. Satellite Derived Bathymetry (SDB)

The methodology of Satellite-Derived Bathymetry (SDB) was developed by Polcyn (1969) [55], who describes a method based on underwater reflectance, underwater optics, and algorithms to derive surveying shallow waters using Earth Observation optical images. With regards to the IHO (International Hydrographic Organization) and the General Bathymetric Chart of the Oceans (GEBCO), Cook Book methodology [56,57] was implemented and modified, importing Sentinel 2 imagery to create SDB datasets near the defined coastline (Figure 7). This approach was implemented in the offshore environment of Thira and Milos using empirically estimated water depth in order to process the images of the aforementioned Sentinel 2 Multispectral Imager (MSI) regarding the 2021 year.

Additionally, Sentinel 2 and Landsat imagery are open source multispectral satellite platforms that may be used for SDB to provide greater detail and resolution. Therefore, the bathymetry determined from optical satellite remote sensing is dependent on the wavelength regarding the absorption coefficient in the water column [58]. Moreover, it is well known that coastal water absorbs less energy than deep water, resulting in an increased reflectance of solar radiation [57].

Following [57], the satellite images were processed in order to extract SDB multispectral information. Existing studies that employ a GEBCO-like workflow often complete the processing stages manually and only apply the workflow to unique satellite images for distinct locations [58–61]. The control depths were scanned and digitized from the Hellenic Military Service topographic sheets of 1/50,000 scale as points in order to be compared with the produced depths of the SDB datasets. The Sentinel 2 mages of 2021 were selected as input, taking into account Level-1C TOA reflectance products for the purpose of utilizing the bands 2,3, and 8, or Blue, Green, and Near-Infrared bands (Figure 8).

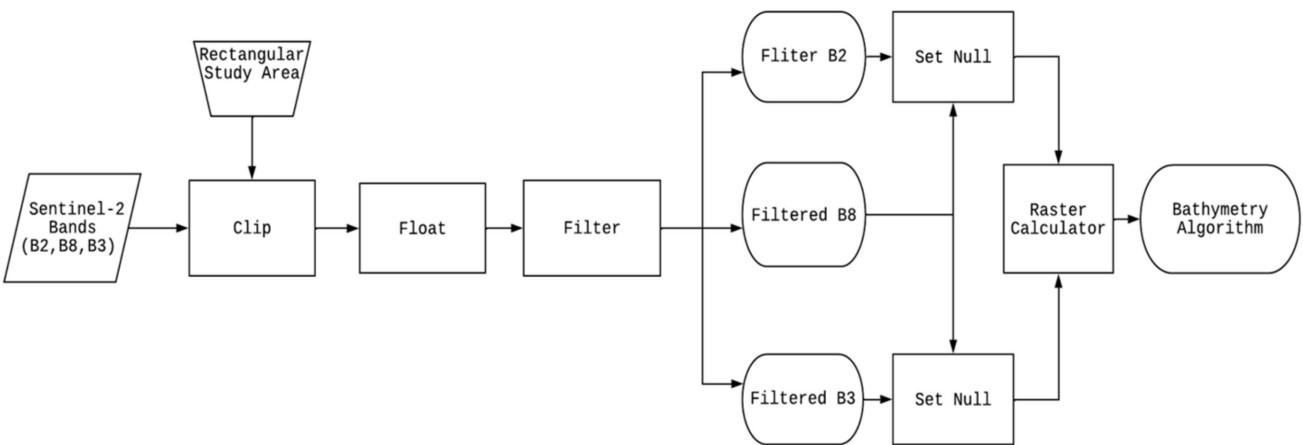

**Figure 8.** Modelbuilder script was developed in this study in order to automatically calculate the bathymetry in shallow waters using three bands (B2, B8, and B3) of Sentinel 2 images.

Another task was the semi-automation of the geoprocessing workflow in the Model-Builder (MB) environment on the ArcGIS software in order to conduct and repeat the empirical procedure on a regional scale for the two islands, as presented in Figure 8. ModelBuilder (MB) is a program for creating, editing, and managing workflows that connect sequences of geoprocessing tools by feeding the result of one tool into the input of another. According to ESRI, MB is an application for developing and executing workflows consisting of a sequence of tools to execute particular processes and Python code to outputs, and it may be utilized in Python scripting and other models. The generated model, in accordance with the GEBCO-like workflow SDB comprises the processing phases depicted in the diagram below.

In relation to the first part of the workflow (Figure 8), all three bands have been converted in floating-point representation of each pixel, which enables the data to support a broad spectrum of digital integers in big decimal representation.

Although most Near-Infrared light is absorbed by clear water, it is still a useful parameter for the land/water boundary extraction process [62]. Green and blue bands penetrate water depths and weaken exponentially as water depth increases, leading to the principle of shallow water bathymetry extraction [48]. Stumpf et al. [63] used a log-band ratio method for SDB mapping that assumes a uniform bottom and a linearly decreasing log-band ratio of water-leaving reflectance with increasing water depth. In the current study, we assumed that water-leaving reflectance contributed to TOA reflectance, and the following Equation (2) for the estimation of SDB:

$$Z = m1 \; X \; \frac{\ln(nRw(\lambda i))}{\ln(nRw(\lambda j))} - m0 \qquad (2)$$

where $Z$ is absolute water depth, $m1$ is a configurable fixed value used to scale ratio to the depth, and $Rw$ (dimensionless) is observed spectral reflectance. For the n-factor we can apply any positive value high enough so that logarithms used in the ratio are positive at every pixel in the image, $m0$ is the offset for the depth of 0 m (z = 0, y-axis intercept), $\lambda i$ is the blue band, and $\lambda j$ is the green band.

Furthermore, the band ratios were calculated following the application of a low-pass three-by-three filter (kernel size of $3 \times 3$). The findings of the band ratios offer a calculation of the relative SDB, which shows the qualitative variation in the water depth in the following section of results (see Section 3.1).

The relative bathymetry becomes unreliable above a certain "extinction depth". Moreover, the extinction depth is critical to determine the reliability of the relative bathymetry. For that purpose, a regression between in situ known bathymetry and relative SDB was implemented. Particularly, a total of 250 points for the two study areas have been digitized and georeferenced in an effort to consider them as in situ water depths. It was observed that more than 150 points from the SDB datasets had very similar values with the charted depths $\pm 1.0$ m accuracy. The following step involved the construction of linear equations in order to convert the SDB values to the water depth data by using the raster calculator within the GIS environment. Above the value of 20 m, it appears that there is only a weak correlation between the derived water depths and the charted depths. In Section 3.1, the resulting equations are shown in the respective scatterplots for the correlation values in accordance to Thira and Milos regions.

### 2.3.3. Random Forest

Random Forest (RF) is a machine learning algorithm, invented by Breiman (2001) [64], which is based on an ensemble of decision trees. The algorithm creates a forest using a big number of individual decision trees trained in parallel with the bootstrap aggregating ensemble technique, generally referred to as Bagging (Figure 9). The RF starts with the selection of random samples generated by small subsets that occurred through the observations of the available features. These subsets are known as bootstrap samples. Bootstrapping ensures that each individual decision tree produced using bootstrap samples is unique, which reduces the overall variance in the RF classifier [65]. The RF classifier is composed by N trees, where N is the number of trees that is produced and specified by the user. Each case of the datasets is passed down to each of the N trees in order to categorize a new dataset [66]. The results of all trees constructed are aggregated, and a vote is taken based on the majority class.

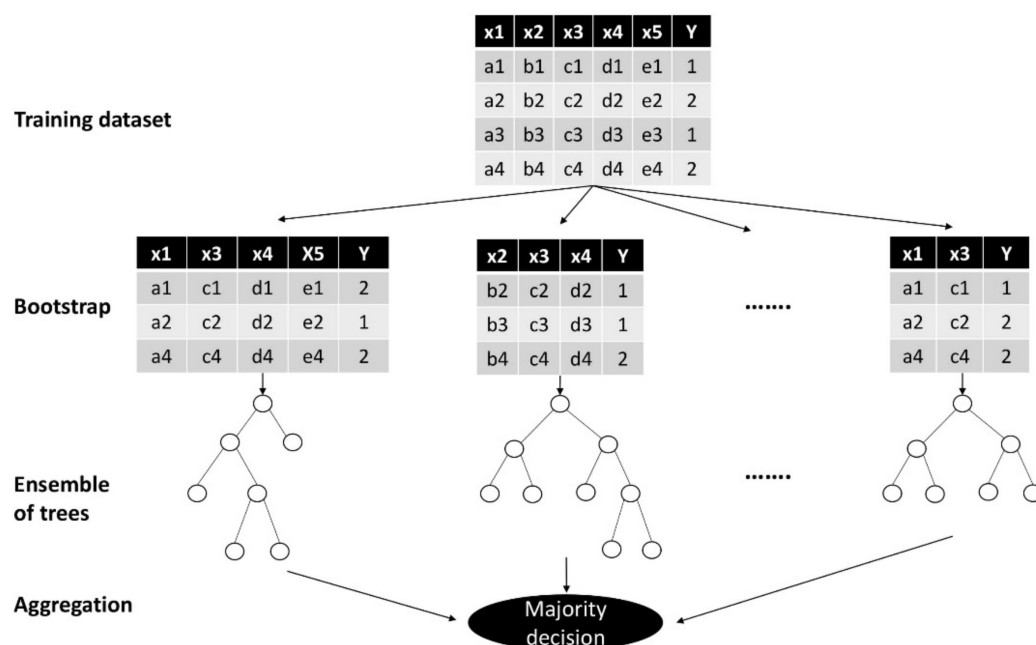

**Figure 9.** The RF classifier is an ensemble method that trains a number of decision trees in parallel with boot-strapping and then aggregates the results. Each tree in the forest is trained using a unique subset of the total training sample and characteristics (modified from [67]).

According to the findings of prior research, the accuracy rate does not considerably improve when the number of trees is increased above 100 [68–70]. The minimum number of trees required to give valid results is 100. However, in this study, the unique architecture of buildings on Cyclades islands, the bare land and the volcanic landscape creates difficulties on the classification separation in terms of spatial identities. RF models can be improved by tuning hyperparameters such as the number of trees, the maximum depth of each decision tree, and the minimum number of samples in a node before the splitting process [71]. In addition, the parametrization of the hyperparameters allow the classifier to adjust the relationships in the produced data set. In Machine Learning (ML) approaches, hyperparameters must be properly fine-tuned to obtain satisfying classification results [72]. In conventional approaches, hyperparameters are usually set empirically or tuned manually [73]. Moreover, very few studies are available for hyperparameter tuning related to land cover classification using satellite images [74]. In this work, tuning of hyperparameters was investigated in order to obtain results that appropriately reflect the quality standards of the study. Specifically, the number of trees was set at 300, the tree depth at 40, and the maximum number of samples was set at 350. The bootstrap samples were collected for Thira and Milos islands (Table 2), using Sentinel-2 images on 5 September 2016 and 18 September 2016, respectively.

**Table 2.** Number of bootstrap samples on every image of 2016.

| Class | Thira | Milos |
|---|---|---|
| Urban | 120 | 110 |
| Non-Urban | 300 | 380 |

### 2.3.4. Classification Accuracy Validation

In an attempt to verify RF classification results, the ground truthing approach implemented using high-resolution images from Google Earth Pro platform. Regarding the accuracy assessment of the RF results statistical indicators were retrieved by analyzing 150 randomly distributed samples from each classified image. These evaluation indicators appear in the confusion matrices, including Overall Accuracy (OA), Producer's Accuracy

(PA), User's Accuracy (UA), and Kappa index (see Section 3.2). The accuracy of each class is described with two different measures referred as user's accuracy, related to errors of omission, respectively [75]. Overall accuracy is also estimated by calculating the sum of the correctly classified sample units divided by the total number of sample units in the created confusion matrix. Another important measurement is Cohen's Kappa proposed by Cohen (1960), which is a robust statistic that can be used for interrater reliability analysis [76]. The K ("Kappa" or "KHAT") statistic is a measure of the difference between the actual agreement between reference data and an automated classifier and the chance agreement between the reference data and a random classifier [77,78]. It was firstly applied in Earth Observation analysis by Congalton and Mead [79] in 1983, and since then, numerous studies [80,81] suggest the technique as one of the most standard component in almost every accuracy assessment [78]. The following Equation (3) is illustrating the computing of K statistic:

$$K = \frac{\sum_{i=1}^{r} X_{ii} - \sum_{i=1}^{r}(X_{i+} \times X_{+i})}{N^2 - \sum_{i=1}^{r}(X_{i+} \times X_{+i})} \tag{3}$$

where $r$ is the number of rows and columns in error matrix, $X_{ii}$ is the number of observations in row i and column. i, $X_{i+}$ is the marginal total of row i, $X_{ii}$ = marginal total of columns. i, and $N$ the total number of observations. The statistical results and the corresponding tables of the classified images are described in Section 3.2.

## 3. Results

### 3.1. Satellite-Derived Bathymetry and Coastal Zone Visualization

According to the literature [58], the Sentinel 2 images were analyzed in order to obtain SDB multispectral data. Regarding the presented methodology, the SDB datasets that were produced using MB environment and the aforementioned Sentinel 2 images contained arithmetic values starting from 0.98 to 1.08 (Figure 10, left image) for Thira and from 1.01 to 1.09 for Milos Island (Figure 10, right image). The produced float values were converted to calculated depths.

This process took place in the Excel program in order to estimate the value of depth extinction regarding the created graphs between the values of charted depth points and measured values (Figure 11). According to the accompanying graphs (Figure 11), the "extinction depth" for both case studies appear to begin at approximately 20 m depth, where the behavior of the linear line begins to curve vertically at deeper water values.

Regarding the statistical analysis performed in the Excel spreadsheet, the following two linear regression equations are presented. In addition, scatterplots were used to convert the estimated values into depths. As gain and offset values, the constraints of the derived equations of the two plots were utilized. Specifically, the gain and offset for Thira were defined as 546.94 and 548.34, whereas for Milos they were 669.38 and 690.63, respectively. In addition, the correlation coefficients ($R^2$) in both regions were calculated as 0.76 for Thira and 0.67 for Milos, taking into account the produced linear equation from the following charts.

Following this process, an isobath with a depth of 10 m was constructed for both of the case studies regarding the selected limit as a critical boundary of the coastal zone's seaward part. Figure 12 illustrates a visual representation of the derived isobath (blue polygon area). In particular, the following maps depict the coastal zone boundaries for both case studies, divided into seaward and landward parts, in an effort to adopt the limits that have been presented in the schematic cross-section in Figure 7.

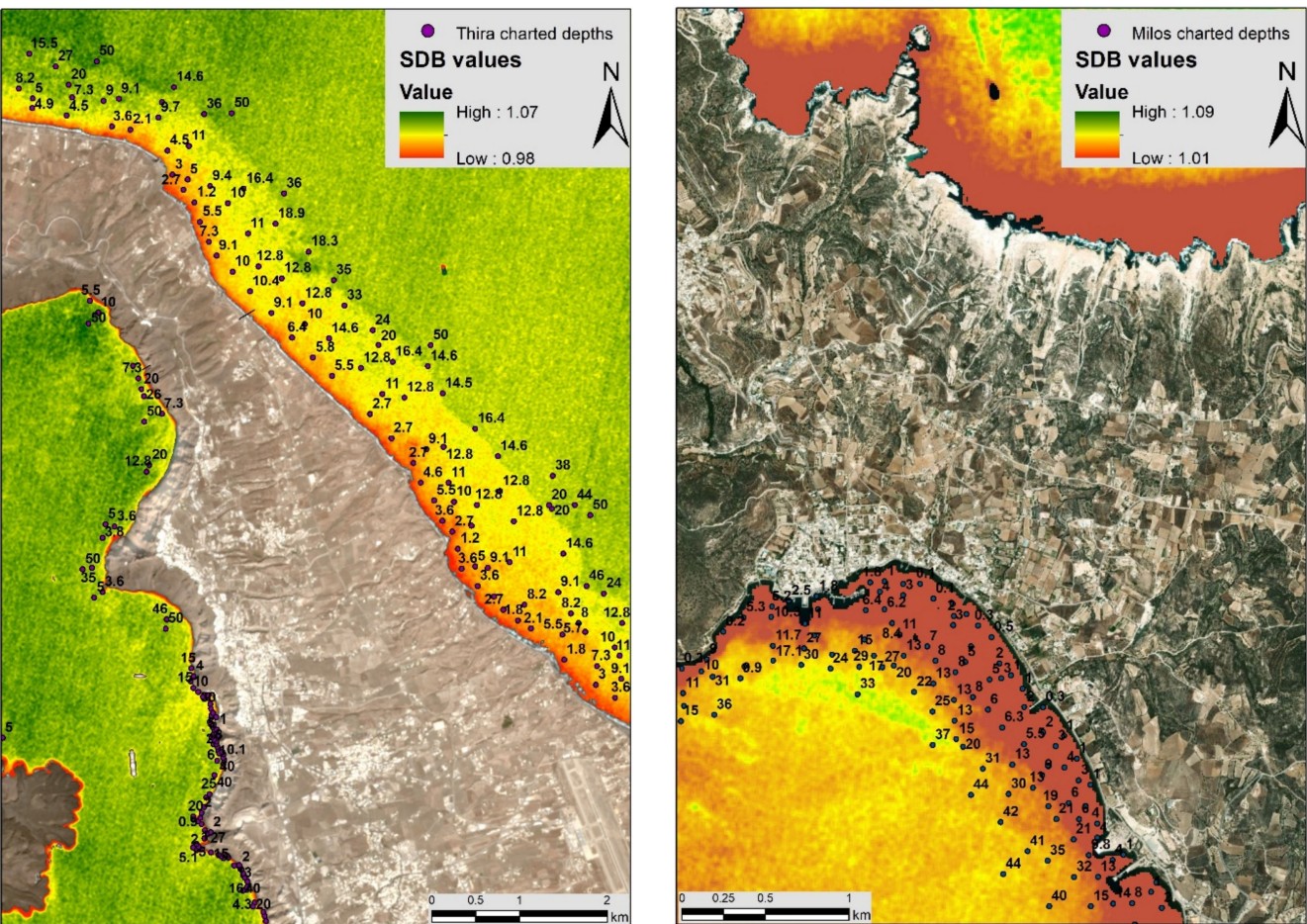

**Figure 10.** SDB algorithm's results focused on the northern part of Thira (**left image**) and centre part Milos (**right image**) using the developed semi-automatic script. Red areas represent shallower depths, in contrast to yellow–green areas that illustrate deeper regions.

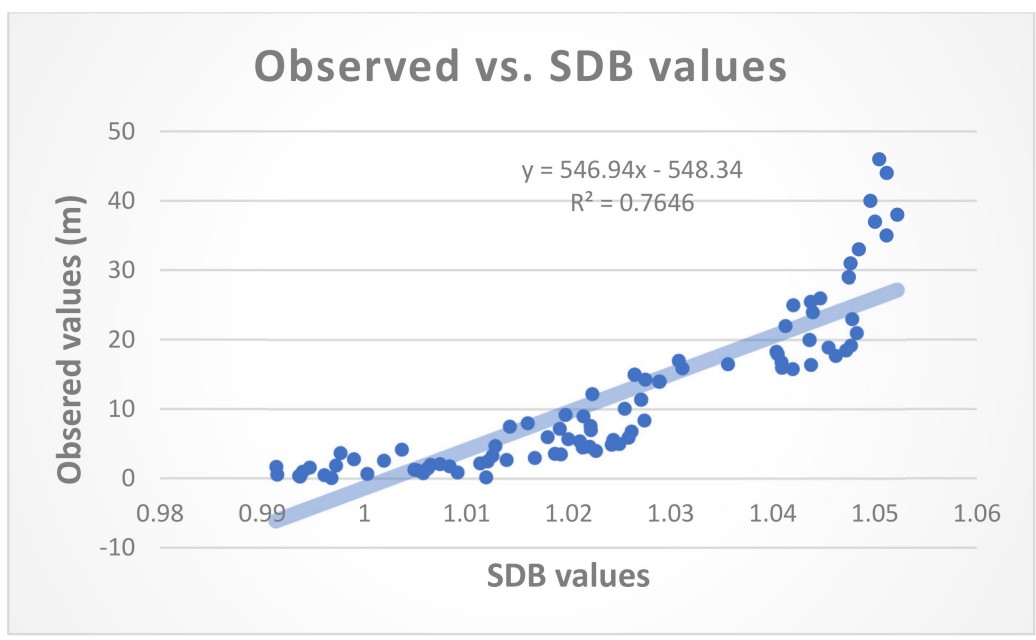

**Figure 11.** *Cont*.

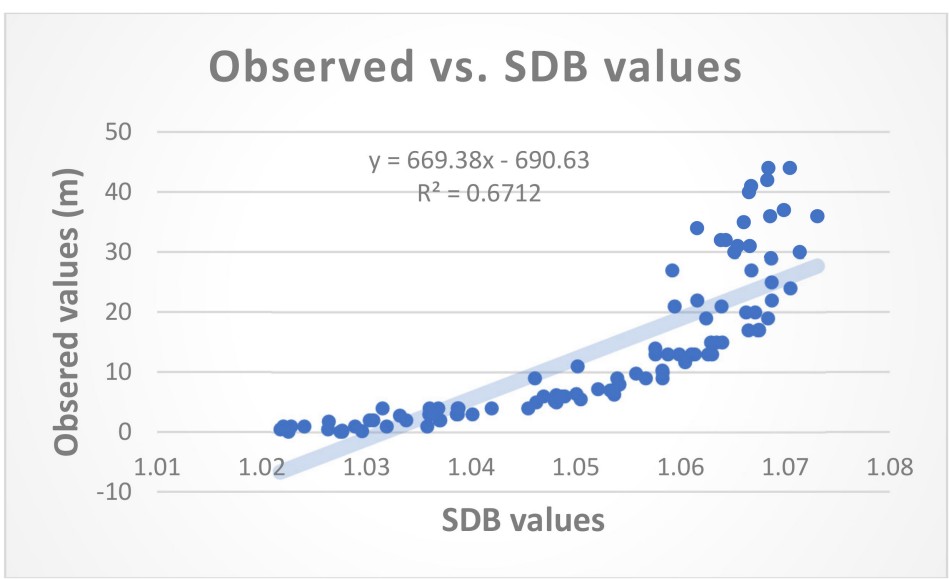

**Figure 11.** Determination of water depth extinction, example for in situ data and SDB from the ln(blue)/ln(green) band ratio, in Thira (top chart) and Milos (bottom chart) islands.

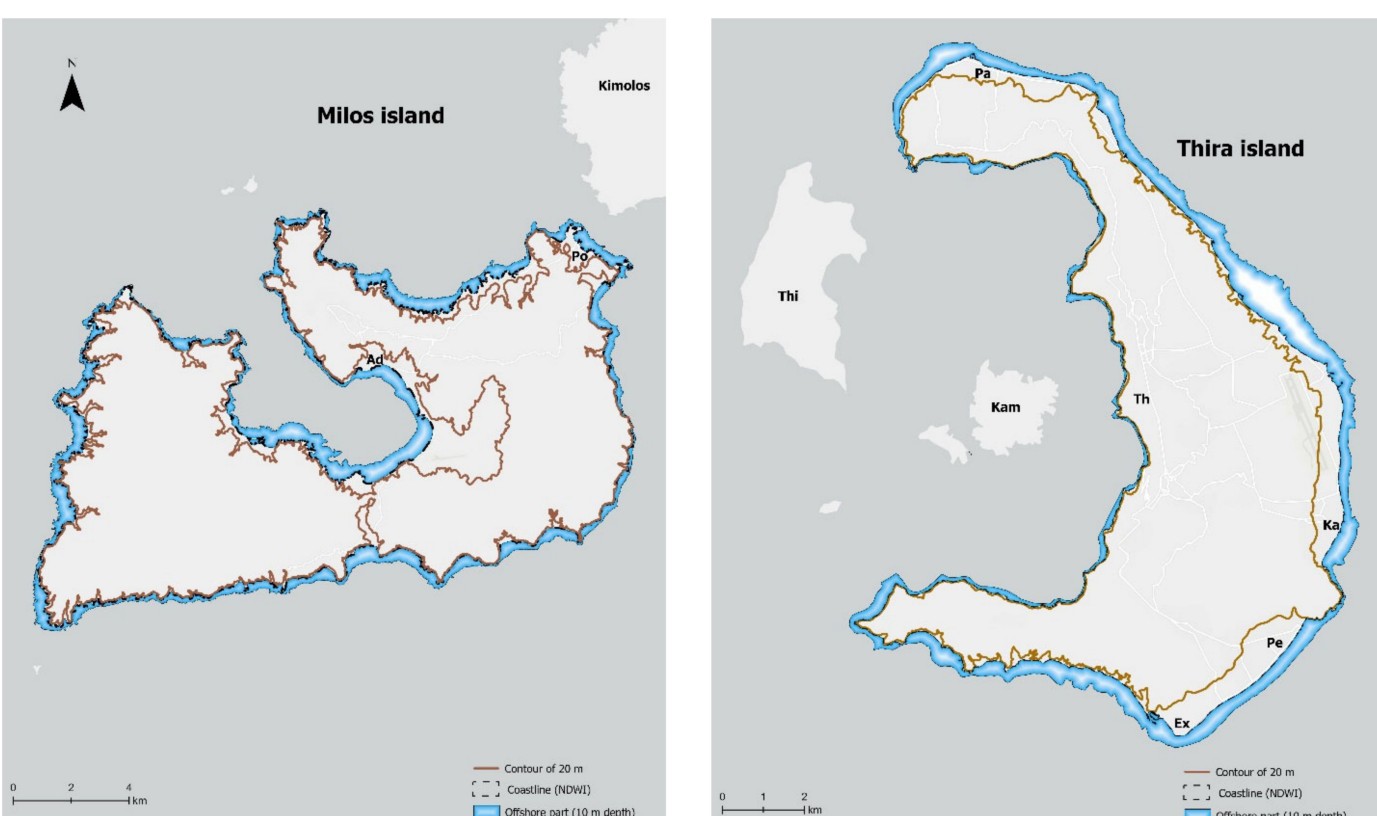

**Figure 12.** The coastal boundaries of Milos (**left**) and Thira (**right**) (Santorini complex) Islands regarding the elevation contour of 20 m (brown line) and the isobath of 10 m (blue polygon). Ad: Adamas, Po: Pollonia, Thi: Thirasia, Kam: Kamenes volcano islands, Ex: Exomitis, Pe: Perissa, Ka: Kamari.

### 3.2. Accuracy Assessment and Validation

To evaluate the accuracy of the results, both qualitative and quantitative evaluations were performed using optical observations and a goodness-of-fit statistical analysis. As mentioned in Section 2.3.3, random sampling was implemented on a regional scale for the accuracy assessment utilizing 150 random points on each classified image. These random samples were utilized to create confusion matrices in order to calculate the evaluation indicators regarding the accuracy of the implemented ML technique. In particular, the columns and the rows in the tables represent the types of the land cover that were obtained from the satellite images using Google Earth Pro and from the applied classification process, respectively. According to Landis [82], the kappa values are divided into six classes in order to maintain consistency in terminology when defining the relative strength of agreement associated with the kappa statistic, which is described as: <0.00, poor; 0.00–0.20, slight; 0.21–0.40, fair; 0.41–0.60, moderate; 0.61–0.80, substantial; and 0.81–1.00, almost perfect.

According to the measurements of the accuracy assessment, it was observed that the highest classification accuracy was performed in the classified image of Thira in 2016 (Table 3), where Kappa reached up to 89.3% and OA scored 94.6%, followed by the classified image of 2021 (Table 4), where Kappa reached 93.3% and OA 87.4%, respectively. On the other hand, the accuracy for Milos had the lowest values regarding Kappa were extended to 82.9% and OA reached up to 91.3% in 2016 (Table 3), followed by the categorized image of 2021 (Table 4) which Kappa reached 77.4% and OA up to 88%.

**Table 3.** Accuracy assessment table for the Thira and Milos classification map (2016).

| Class | Urban | Non-Urban | Producer Acc. | User Acc. |
|---|---|---|---|---|
| Urban (Thira) | 75 | 6 | 97.4% | 92.5% |
| Non-Urban (Thira) | 2 | 67 | 91.7% | 97.1% |
| Thira Overall Acc: 94.6%; Kappa Statistic: 89.3% | | | | |
| Urban (Milos) | 49 | 6 | 87.5% | 89% |
| Non-Urban (Milos) | 7 | 88 | 93.6% | 92.6% |
| Milos Overall Acc: 91.3%; Kappa Statistic: 82.9% | | | | |

**Table 4.** Accuracy assessment table for the Thira and Milos classification map (2021).

| Class | Urban | Non-Urban | Producer Acc. | User Acc. |
|---|---|---|---|---|
| Urban (Thira) | 74 | 5 | 93.6% | 93.6% |
| Non-Urban (Thira) | 5 | 66 | 92.9% | 92.9% |
| Thira Overall Acc: 93.3%; Kappa Statistic: 87.4% | | | | |
| Urban (Milos) | 50 | 5 | 79.3% | 90.9% |
| Non-Urban (Milos) | 13 | 82 | 94.2% | 86.3% |
| Milos Overall Acc: 88%; Kappa Statistic: 77.4% | | | | |

Taking into account the lower values on Milos, the analysis of UA and PA (Tables 3 and 4) measurements illustrates that the highest rates of misclassification are detected in the Urban class. In order to investigate the causes that affect the accuracy of the results, visual observations on the island depict that the most classification errors are highlighted within the boundaries of mine sites, which are categorized as urban areas. For that purpose, specific samples were selected in mining areas in order to compare the spectral signature behavior between the urban and the misclassified regions. The analysis of this comparison (Figure 13) showed that the distribution of the corresponding spectral values is similar causing difficulties in the classification outputs. In the following Figure, 11 bands of the visible and specifically Blue (2), Green (3), Red (4), near-infrared (NIR/8), and short-wave infrared (SWIR/11) channels are presented. In accordance with the generated signature plot (Figure 13), there is a high similarity between the spectral average values of open-pit mines and urban classes in terms of bands set, with higher variances in bands 7 and 9.

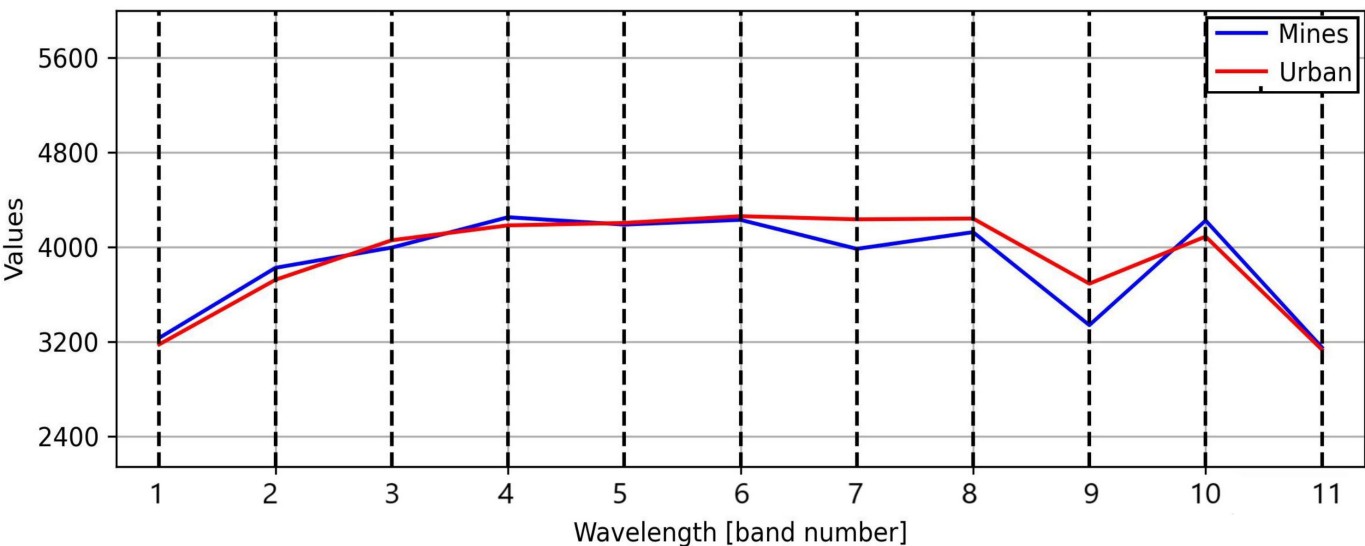

**Figure 13.** Spectral signature plot for the category of open-pit mines (blue line) and urban areas (red line) on Milos island as obtained by Sentinel-2 image in 2016. Each line reflects the dependence of the dimensionless surface reflectance value (vertical axis) on the corresponding band set wavelength (horizontal axis).

Classification products were validated by comparing 2016-created results with ESM and high-spatial-resolution Google Earth Pro (GEP) images. In the instance of Thira, ESM and GEP comparisons reveal a high level of consistency (Table 5). More specifically, the ESM dataset spans over the years 2014–2016. Our extracted urban area, which is based on an RF algorithm, was generated using Sentinel-2 images from September 2016 in order to account for prospective urban extensions that may have occurred between 2014 and 2016.

**Table 5.** Comparison of ESM with ML product for Thira for 2016 and its spatial expansion to 2021.

| Dataset | Area (m$^2$) |
|---|---|
| ESM | 4,460,276 |
| Santorini 2016 | 4,863,371 |
| Santorini 2021 | 4,903,105 |

Considering the fact that ESM datasets are based on data with very high spatial resolution (over 2 m), they are capable of distinguishing buildings from road networks. On the contrary, due to the lower spatial resolution of Sentinel-2 products, it is difficult to isolate the road network from the continuous urban fabric. According to this line of thinking, the major road network was obtained from OpenStreetMap in order to erase it from the built-up area.

In contrast, the results of Milos were more accurate than the ESM dataset on the island's central-north part. Figure 14 illustrates indicative areas in Milos that were misclassified as built-up areas based on the existing ESM dataset (Figure 14a,c), but the resulting RF products demonstrated a clear distinction between the urban environment and the bare land (Figure 14b,d).

In contrast to the well-trained RF dataset, the ESM failures may be related to distinctive volcanological landforms such as the ones at Sarakiniko beach (Figure 14 a,b) and in other areas where the bright color of the surface leads to incorrect classification, resulting in categorization as urban regions.

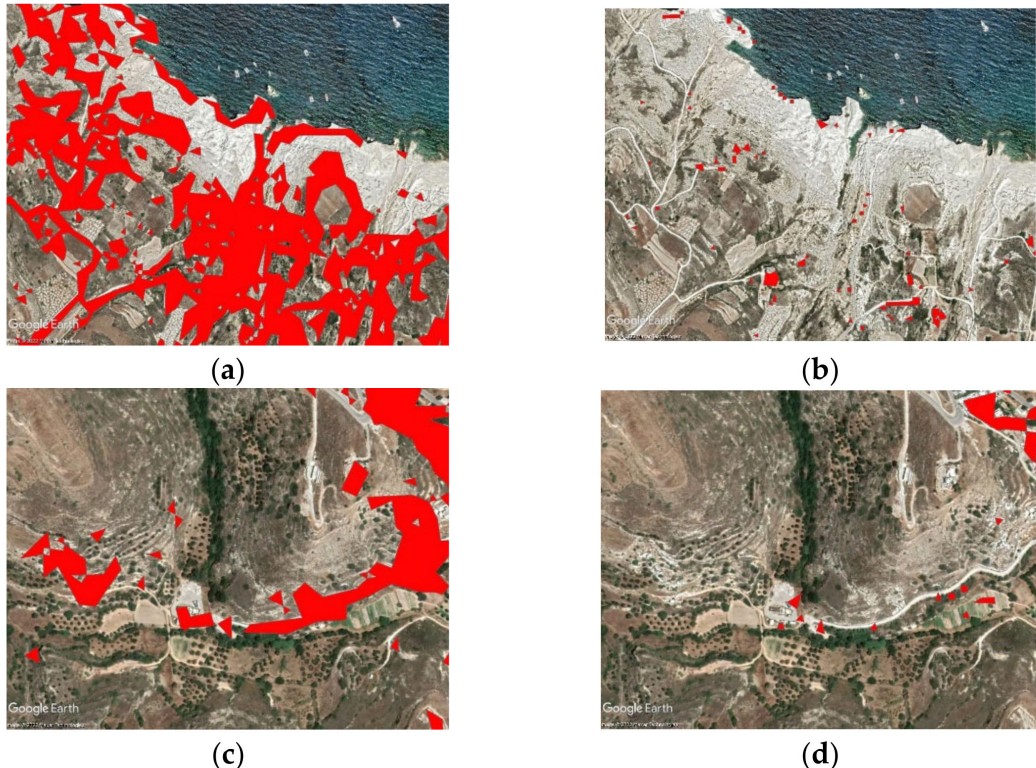

**Figure 14.** The images illustrating the indicative areas in Milos that were misclassified as built-up areas based on the existing ESM dataset (**a**,**c**), in contrast with the resulting RF products demonstrated a clear distinction between the urban environment and the bare land (**b**,**d**).

*3.3. Random Forest Quantification*

The RF algorithm was applied for the years of 2016 and 2021. The urban environment was extracted on regional scale and analyzed within the boundaries of the adopted onshore 20 m elevation limit. The areas included in this elevation range are regarded as E-LECZ [17], and they were processed to determine the urban extension over the five-year period. On Thira, the higher urbanization rates are detected in the southeast part of the island within the boundaries of the selected onshore limit. More precisely, in the Perissa settlement (Figure 15), an urban extension of up to 25.71% (Table 6) was detected, while the highest rates were detected in the Exomitis settlement (Figure 15), which reached up to 43.67%. It should be noted that the spatial resolution of Sentinel-2 affects the detection accuracy of small buildings whose width and length are lower than 10 m. There are cases of existing buildings that were not detected in 2016 due to their extent and appeared in 2021 due to their small expansions that reached or exceeded the spatial resolution of Sentinel-2, and as a result, they were detected as urban classes. Furthermore, spectral similarities and the reflection of solar radiation affect the quantification of the results, showing small extensions while the qualitive images clearly depict the greater changes. A characteristic example is the Kamari settlement, which has an extension of up to 1.6% (Table 6), while according to Figure 15, the changes should be expected to be greater.

**Table 6.** Santorini's coastal settlements evolution for the time period 2016–2021.

| Settlement | Area (m$^2$) 2016 | Area (m$^2$) 2021 | Difference | Urban Expansion (%) |
|---|---|---|---|---|
| Exomitis | 11,625 | 16,701.62 | +5076.62 | 43.67 |
| Perissa | 293,540 | 368,996.3 | +75,456.3 | 25.71 |
| Kamari | 313,453.9 | 318,511.3 | +5057.39 | 1.61 |

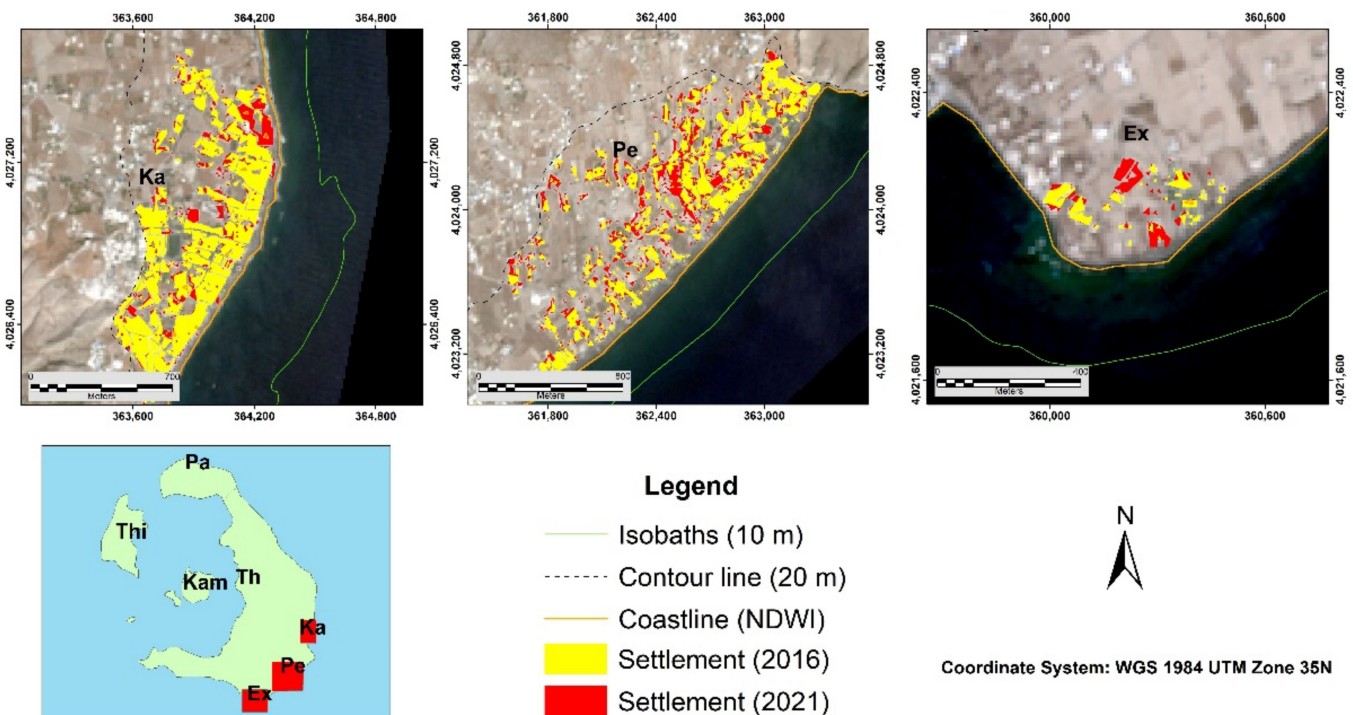

**Figure 15.** The map is illustrating the urban changes during the period 2016–2021 in Kamari, Perissa, and Exomiti settlements. Yellow- and red-color polygons represent the urban identification in 2016 and 2021. The orange line is the coastline derived from NDWI, the green line is the isobath (10 m) extracted by the SDB, and the black dashed lines are the upper limit of the landward region regarding the elevation of 20 m.

On Milos Island, the results depict that there is a relative stable rate of urbanization for the time period of 2016 to 2021. The higher urbanization rates are detected in the central and northeast part of the island. More specifically, in Adamas settlement (Figure 16), an urban extension of up to 36.53% (Table 7), was identified, while lower rates were detected in the Pollonia settlement (Figure 16), reaching up to 36.53%. The most remarkable thing concerning the results of Milos is related to the higher accuracy compared with the ESM 2015 dataset. Due to the unique combination of urban environment and bareland lithology in Milos, there are many locations that are misclassified as urban environments on the ESM dataset while the RF algorithm has successfully isolated these types of failures. However, both products are facing difficulties in the separation of mineral extraction sites from built-up areas. This problem is related to the intensity of the reflected radiation that seems to be very close or similar in both types of spatial identities and, as a result, the classification fails. In order to eliminate this problem, mineral extraction sites were excluded, using as a mask the homonymous category of CLC 2018.

**Table 7.** Milos's coastal settlements evolution for the time period 2016–2021.

| Settlement | Area (m²) 2016 | Area (m²) 2021 | Difference | Urban Expansion (%) |
|---|---|---|---|---|
| Pollonia | 79,296 | 108,264 | +28,967 | 36.53 |
| Adamas | 344,410 | 395,266 | +50,856 | 14.77 |

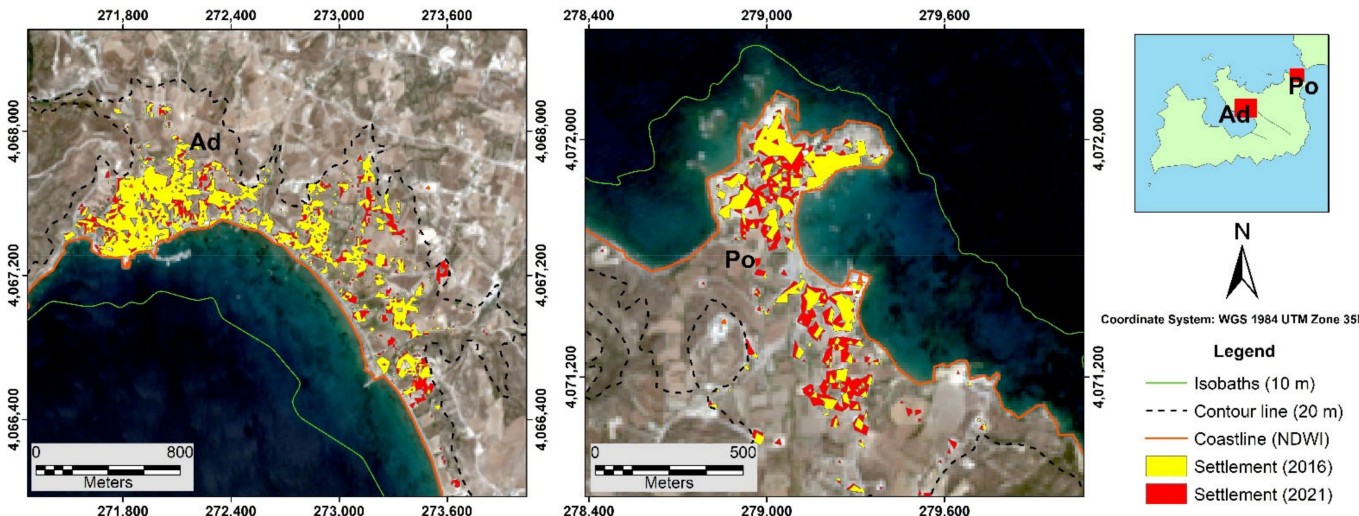

**Figure 16.** The map is illustrating the urban changes during the period 2016–2021 in Pollonia and Adamas settlements. The orange line is the coastline derived from NDWI, the green line is the isobath (10 m) extracted by the SDB, and the black dashed lines are the upper limit of the landward region regarding the elevation of 20 m.

## 4. Discussion

Monitoring urban development enables researchers and policymakers to deal with the consequences of specific actions e.g., in the context of urban planning, or to better envisage future inventions that can minimize the potential exposure to hazards such as coastal flooding, tsunami, and coastal erosion. The spatially results in terms of urban extent and coastal zone boundaries can provide the possibility to analyze different urban features and designate future priority areas for sustainable and resilient coastal planning in the South Aegean islands.

In this frame, this work tries to determine the coastal zone borders combining published reports and relative literature in islands of Milos and Thira (Santorini Island complex) of the South Aegean Volcanic Arc. Additionally, the Aegean Sea coastal zone is an important part of Greece, as it represents the majority of the country's socioeconomic and industrial endeavors. As previously stated, more than a third of Greece's population and over than 85% of the country's economic activity, predominantly tourism, reside within a few kilometers of this specific coastal region. Moreover, urbanization expansion and climate change pose a threat to the natural environment, due to the modification of the physical land, especially in low-lying coastal areas. The continuous loss of coastal land combined with Extreme Sea Levels (ESLs) scenarios in Europe predict an increase in sea level of 1 m or more by the end of the century, which will result in a worsening of the negative impacts in the coastal zone.

Despite the fact that the geographical limits of the coastal units have been examined in prior research [29] and EU-funded programs, it still remains a complex and multi-parametric issue. The evolution of climate change, the topographical characteristics, the morphology of the unified onshore and offshore landscape, and the urban features on the landward part play a key role in the coastal planning strategy and climate change adaptation.

The study also presented a GEOINT approach, combining satellite images and geospatial information in an effort to analyze the human footprint and its evolution on the defined coastal areas within a specific spatiotemporal environment from the 2016 to 2021 time period. The study also highlighted inaccuracies in existing open-source datasets, due the complexity of the unique Cycladic architectural style together with bare-soil characteristics of the areas under examination.

Finally, RF as an ML methodology is an acceptable approach that needs continuous development based on the parameterization of the hyperparameters and spatial resolution of processed satellite images. Furthermore, the SDB methodology showed adequate results until the depth of around 20 m, having a good correlation between the charted and the predicted depths. However, the implemented SDB approach is an empirical process and could be optimized with the utilization of very high-resolution images, more reliable ground truth measurements, and more accurate control points. In addition, the correlation coefficient ($R^2$) in both case studies was not very high due to several sources of errors. A possible cause could be the accuracy of the georeferenced bathymetric maps, the turbidity and the spatial resolution of the processed satellite images [83].

Our work shows that the average accuracy of the applied methodology for urban mapping using RF in a GIS environment achieved very good results in terms of having a Kappa index higher than 80%. Additionally, the outcome illustrates that the average urbanization expansion within the boundaries of selected settlements was greater than 20% for both case studies. The findings could contribute to the effective and holistic management on similar coastal environment in the context of climate change adaptation, mitigation strategies, and multi-hazard assessment globally.

## 5. Conclusions

It is noteworthy that one of the most important parameters regarding the vulnerability and exposure of coastal areas in Aegean Sea are the ever-increasing urbanization expansion in places and the climate change trend. In accordance with the findings, existing spatial datasets of urban expansion are accessible at high resolution and at local or regional geographical scales, limiting their usage due to their misclassification and near real-time update.

In this study, we introduced an innovative workflow in an effort to (i) delineate coastal bounds, (ii) assess depths values near coastline, and (iii) monitor urban patterns in volcanic islands of the South Aegean region. In an attempt to visualize the urban expansion, RF algorithm was implemented as a semi-automatic classification approach adjusted on the unique characteristics of Thira and Milos islands. The findings showed that in specific settlements the average increasing rate of urbanization was more than 20% from 2016 to 2021. The results related to the urban growth, especially in low-lying coastal areas, might result in substantial negative changes in assessing future coastal urban exposure and management.

The depicted geospatial workflow can efficiently monitor urbanization mapping and shallow depths near the coastline in order to support effective strategy and policies to potential coastal risk, depending on the selected SLR scenario, the climate change, and the potential multi-hazards. Additionally, the geographical delineation of the presented coastal unit could be useful for civil protection, regional planning, decision making, and a early multi-hazard warning system. Thus, due to the high tourism congestion on these two specific islands the output could support a better quantification of the existing open spaces and beaches in an effort to provide a more thorough planning against COVID-19 and other pandemic restrictions that require safe distances and suitable open spaces.

Summarizing, this study also underlines the significance of using remote sensing and geoinformation techniques to study coastal areas, not only on a regional but even more on a local scale in terms of GEOINT. This work also investigates coastal environments based on different scenarios and assumptions. Future research could be based on very high spatial resolution satellite and/or aerial images in order to optimize results from the SDB (e.g., correlation coefficient $R^2$) and machine learning classification techniques. The findings of the urbanization expansion could be correlated with existing land use plannings and/or can be utilized to predict future population density changes. An upcoming work will be the development of a Multiple Criteria Decision Making (MCDM) approach in the same period of time (2016 to 2021) in an attempt to identify the most vulnerable areas within the boundaries of the unified onshore and offshore environment.

**Author Contributions:** Conceptualization, P.K., A.K. and I.P.; methodology, P.K. and A.K.; validation, P.K., N.K., P.N. and S.K.; formal analysis, P.K., K.K. and N.K.; investigation, P.K., A.K., K.K. and N.K.; resources, P.K. and N.K; data curation, P.K., A.K., K.K. and P.N.; writing—original draft preparation, P.K.; writing—review and editing P.K., A.K., P.N., K.K., N.K., I.P. and S.K.; visualization, P.K. and A.K; supervision P.K. and N.K. All authors have read and agreed to the published version of the manuscript.

**Funding:** This research received no external funding.

**Institutional Review Board Statement:** Not applicable.

**Informed Consent Statement:** All individuals included in this section have consented to the acknowledgement.

**Data Availability Statement:** Data available upon request.

**Acknowledgments:** Maps and diagrams throughout this work were created using ArcGIS® software by Esri. ArcGIS® and ArcMap™ are the intellectual property of Esri and are used herein under license. Copyright © Esri. All rights reserved. For more information about Esri® software, please visit https://www.esri.com/en-us/home (accessed on 3 August 2022). Many thanks are also given to the colleagues from CERTH Evangelia Zygouri and Kleomenis Kalogeropoulos from ELSTAT, for their participation in the editing and data information. The authors are grateful to the European Space Agency and the National Aeronautics and Space Administration, who provided Sentinel-2 and SRTM data accordingly. The authors would also like to thank the reviewers for providing useful suggestions that enhance the manuscript's quality.

**Conflicts of Interest:** The authors declare no conflict of interest.

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
