# Peer review of "Geospatial Intelligence and Machine Learning Technique for Urban Mapping in Coastal Regions of South Aegean Volcanic Arc Islands"

_2673-7418, doi:10.3390/geomatics2030017_

Round 1

Reviewer 1 Report

For the most part, the authors answered my questions and followed the recommendations. Otherwise, I wanted the authors to go a little deeper into the problem of urbanization itself or to highlight aspects of spatial planning at the sea-land interface (lines 50-132) that maritime countries face. I also suggested some references along these lines, but OK.

Now I have no further comments and consider the article suitable for publication.

Author Response

Dear reviewer,

We appreciate the opportunity to submit the revised version of the paper "Geospatial Intelligence and Machine Learning Technique for Urban Mapping in Coastal Regions of South Aegean Volcanic Arc Islands" for consideration by the Geomatics Journal (MDPI). We appreciate the resources and your valuable time you invested in offering important comments on our manuscript. Your notifications and corrections were very helpful to add value in our work and to make the necessary corrections and changes in the whole manuscript.

Reviewer 1

Comments and Suggestions for Authors

For the most part, the authors answered my questions and followed the recommendations. Otherwise, I wanted the authors to go a little deeper into the problem of urbanization itself or to highlight aspects of spatial planning at the sea-land interface (lines 50-132) that maritime countries face. I also suggested some references along these lines, but OK.

Now I have no further comments and consider the article suitable for publication.

Author’s response: Dear reviewer, thank you for your response. We are happy to know that we have answered your questions. Indeed, future work will be carried out to improve ML techniques and to further investigate spatial planning related to coastal regions.

Reviewer 2 Report

I am OK with the answers the authors provided to my comments in the rebuttal letter and have no further comments. 

Author Response

Dear reviewer,

We appreciate the opportunity to submit the revised version of the paper "Geospatial Intelligence and Machine Learning Technique for Urban Mapping in Coastal Regions of South Aegean Volcanic Arc Islands" for consideration by the Geomatics Journal (MDPI). We appreciate the resources and your valuable time you invested in offering important comments on our manuscript. Your notifications and corrections were very helpful to add value in our work and to make the necessary corrections and changes in the whole manuscript.

Comments and Suggestions for Authors

I am OK with the answers the authors provided to my comments in the rebuttal letter and have no further comments. 

Author’s response: Dear reviewer, thank you for your reply. It is very important the fact that we have managed to answer your comments in an effort to improve our manuscript.

Reviewer 3 Report

Thank you to the authors for the extensive revisions to the manuscript. In most aspects I am satisfied with the quality of the updated paper. The authors addressed all points more or less adequately. The abstract has a much better length now and the Discussion and Conclusions provide more detail.

However, the manuscript still requires extensive English language editing by a qualified native speaker. There are numerous grammatical errors (the below examples are only an indication of this, there are more which I have not marked). This distracts from the actual content of the paper.

The paper states that specific literature is used to determine the South Aegean coastline boundaries. However apart from a few reports, there are hardly any references to any literature at all so this is very misleading. There is no critical elaboration on the literature being used.

Some specific comments:

Line 16-17: The first sentence needs to be revised, it is not logical. Also, provide examples of “coastal enterprises”.

Line 20: No comma after GEOINT

Line 22: What is sustainable delineation?

Line 23f.: no “in”. Also, use commas instead of semicolons.

Line 41: specify “close”. Distance, time to travel by car? How much?

Line 45: vital and important - seems redundant. Also in the following sentence: “critical” and “high level of importance”.

Line 49: sixth biggest destination: what is the source of this information?

Line 59: no comma after wetlands

Line 76: … and describes… (missing “and”)

Line 78ff.: please revise this sentence.

Line 86f.: this paper illustrates… The continuous form is used frequently throughout this paper, this should be corrected.

Line 107: introduction (not introductory) - (or just: as stated above)

Line 115f.: inconsistent use of comma separator

Lines 107-143: This section provides important additional information that was missing before. However it still needs English language editing. The same applies to the Discussion and Conclusions sections.

Line 170: please specify “historical times”

Line 179: it is still not clear from this sentence that Thira is a part of the Santorini complex (and that Santorini is a complex and not just a single island).

Line 180f.: revise this sentence; Thira is an island and includes other islands? Something is not right here. 

Line 181: Christiana (missing h)

Line 197: “section” would fit better than “manuscript”

Line 212: “moreover” doesn’t fit here either. Extensive English revision required.

Line 217-222: revise. Missing verb, incomplete sentences.

Line 257: how can Boolean logic be applied by creating a cloud mask? Isn’t the cloud mask the result of this logic rather than a means to create it?

Line 264: the image represents…

Line 280: rephrase (the implementation… was implemented)

Line 285f.: rephrase

Line 324-326: revise sentence - missing verb

Line 348: Additionally (typographic error). This sentence reads as if Sentinel and Landsat are commercial platforms. Please rephrase.

Line 379: the term “thesis” is still used in this manuscript

Line 422: Bagging

Line 580: check position of >. It is also not clear where the 2m resolution comes from at this stage; this needs to be detailed in the Data section. My understanding is that ESM is based on 2.5m SPOT 5/6 sensors and the final product has a resolution of 2 or 10m. It is mentioned earlier that this study uses the 10m imagery. More clarification on this would help the reader to understand this better.

Line 584: this is unclear. The road network was extracted from OSM and then subtracted from the ESM??? This sentence implies something different.

Line 449, 591, 596, 634: RT is not explained in this manuscript. The authors intended to only use RF but this has not been carried out sufficiently.

Line 650: “therefore” doesn’t make sense here

Line 662-665: revise. Grammatically incorrect.

Line 673: literation???

Author Response

Dear reviewer,

We appreciate the opportunity to submit the revised version of the paper "Geospatial Intelligence and Machine Learning Technique for Urban Mapping in Coastal Regions of South Aegean Volcanic Arc Islands" for consideration by the Geomatics Journal (MDPI). We appreciate the resources and your valuable time you invested in offering important comments on our manuscript. Your notifications and corrections were very helpful to add value in our work and to make the necessary corrections and changes in the whole manuscript. For a detailed response to your  comments and concerns, please read the section below.

Reviewer 3

Comments and Suggestions for Authors

Thank you to the authors for the extensive revisions to the manuscript. In most aspects I am satisfied with the quality of the updated paper. The authors addressed all points more or less adequately. The abstract has a much better length now and the Discussion and Conclusions provide more detail.

However, the manuscript still requires extensive English language editing by a qualified native speaker. There are numerous grammatical errors (the below examples are only an indication of this, there are more which I have not marked). This distracts from the actual content of the paper.

The paper states that specific literature is used to determine the South Aegean coastline boundaries. However apart from a few reports, there are hardly any references to any literature at all so this is very misleading. There is no critical elaboration on the literature being used.

Some specific comments:

Line 16-17: The first sentence needs to be revised, it is not logical. Also, provide examples of “coastal enterprises”.

Author’s response: Revised accordingly.

Line 20: No comma after GEOINT

Author’s response: Revised accordingly.

Line 22: What is sustainable delineation?

Author’s response: Revised accordingly. In general, this work tries to delineate the coastline boundaries of the South Aegean islands.

Line 23f.: no “in”. Also, use commas instead of semicolons.

Author’s response: Revised accordingly.

Line 41: specify “close”. Distance, time to travel by car? How much?

Author’s response: Thank you for the comment, the specific sentence revised following your to advice in terms of distance.

Line 45: vital and important - seems redundant. Also in the following sentence: “critical” and “high level of importance”.

Author’s response: Revised accordingly. 

Line 49: sixth biggest destination: what is the source of this information?

Author’s response: Thank you very much for your notice, reference has been added.

Line 59: no comma after wetlands

Author’s response: Revised accordingly.

Line 76: … and describes… (missing “and”)

Author’s response: Revised accordingly.

Line 78ff.: please revise this sentence.

Author’s response: Revised accordingly. Thank you very much for the comment, the text has been revised and shortened in an attempt to be more meaningful and understandable.

Line 86f.: this paper illustrates… The continuous form is used frequently throughout this paper, this should be corrected.

Author’s response: Revised accordingly.

Line 107: introduction (not introductory) - (or just: as stated above)

Author’s response: Revised accordingly.

Line 115f.: inconsistent use of comma separator

Author’s response: Revised accordingly.

Lines 107-143: This section provides important additional information that was missing before. However, it still needs English language editing. The same applies to the Discussion and Conclusions sections.

Author’s response: Thank you very much for the comment, the English language has been edited in an attempt to be more simple and understandable.

Line 170: please specify “historical times”

Author’s response: Dear reviewer thank you for your comment. Related reference has been added to specify the time period of “historical times”.

Line 179: it is still not clear from this sentence that Thira is a part of the Santorini complex (and that Santorini is a complex and not just a single island).

Author’s response: Revised accordingly.

Line 180f.: revise this sentence; Thira is an island and includes other islands? Something is not right here.

Author’s response: Revised accordingly. Thira is the largest island of Santorini complex.

Line 181: Christiana (missing h)

Author’s response: Removed.

Line 197: “section” would fit better than “manuscript”

Author’s response: Revised accordingly.

Line 212: “moreover” doesn’t fit here either. Extensive English revision required.

Author’s response: Revised accordingly.

Line 217-222: revise. Missing verb, incomplete sentences.

Author’s response: Revised accordingly.

Line 257: how can Boolean logic be applied by creating a cloud mask? Isn’t the cloud mask the result of this logic rather than a means to create it?

Author’s response: Thank you for pointing this out. Indeed, the cloud mask is applied to separate the images on cloud and non-cloud areas so the Boolean logic is removed from the manuscript.

Line 264: the image represents…

Author’s response: Revised accordingly.

Line 280: rephrase (the implementation… was implemented)

Author’s response: Rephased accordingly.

Line 285f.: rephrase

Author’s response: Revised accordingly.

Line 324-326: revise sentence - missing verb

Author’s response: Revised accordingly.

Line 348: Additionally (typographic error). This sentence reads as if Sentinel and Landsat are commercial platforms. Please rephrase.

Author’s response: Revised accordingly.

Line 379: the term “thesis” is still used in this manuscript

Author’s response: Revised accordingly.

Line 422: Bagging

Author’s response: Revised accordingly.

Line 580: check position of >. It is also not clear where the 2m resolution comes from at this stage; this needs to be detailed in the Data section. My understanding is that ESM is based on 2.5m SPOT 5/6 sensors and the final product has a resolution of 2 or 10m. It is mentioned earlier that this study uses the 10m imagery. More clarification on this would help the reader to understand this better.

Author’s response: Indeed, the spatial resolution of ESM products has 2 and/or 10 m. However, the products were generated using high spatial resolution images like SPOT 5/6, Pleiades and Worldview (0,45 m to 2,5 m), and as result, they are very detailed in terms of spatial information. On the other hand, the Sentinel-2 products are based on 10 m pixel cell size missing variable information compared with the ESM data inputs.

Line 584: this is unclear. The road network was extracted from OSM and then subtracted from the ESM??? This sentence implies something different.

Author’s response: Due to the very high spatial resolution of ESM products, the road network is already separated from buildings. In case of the classified Sentinel-2 images due to their spatial resolution at 10 m it is difficult to separate the roads from built-up areas. In order to manage this issue, the major road network was derived from OSM portal and then was erased from the classified machine learning product.

Line 449, 591, 596, 634: RT is not explained in this manuscript. The authors intended to only use RF but this has not been carried out sufficiently.

Author’s response: Thank you for highlight this. We check the manuscript and make all the necessary changes.

Line 650: “therefore” doesn’t make sense here

Author’s response: Has been removed.

Line 662-665: revise. Grammatically incorrect.

Author’s response: Revised accordingly.

Line 673: literation???

Author’s response: Revised accordingly

Reviewer 4 Report

I carefully studied the submission titled "Geospatial Intelligence and Machine Learning Technique for Urban Mapping in Coastal Regions of South Aegean Volcanic Arc Islands". Thank you for the invitation. As a result of my elaborations, I think that the article is not ready for publication in its current form and requires a serious revision. I am attaching the letter to which I presented my concerns.

Author Response

Dear reviewer,

We appreciate the opportunity to submit the revised version of the paper "Geospatial Intelligence and Machine Learning Technique for Urban Mapping in Coastal Regions of South Aegean Volcanic Arc Islands" for consideration by the Geomatics Journal (MDPI). We appreciate the resources and your valuable time you invested in offering important comments on our manuscript. Your notifications and corrections were very helpful to add value in our work and to make the necessary corrections and changes in the whole manuscript. For a detailed response to your comments and concerns, please read the section below.

Reviewer 4

Comments and Suggestions for Authors

I carefully studied the submission titled "Geospatial Intelligence and Machine Learning Technique for Urban Mapping in Coastal Regions of South Aegean Volcanic Arc Islands". Thank you for the invitation. As a result of my elaborations, I think that the article is not ready for publication in its current form and requires a serious revision. I am attaching the letter to which I presented my

Issue 1: There are serious problems in the submission in terms of the use of terminology, word choice, grammar, correct use of abbreviations, and presentation of quantitative results.

- Abbreviations are given without their long forms. Some terms are abbreviated but not reused.

Author’s response: Revised accordingly.

- In Line 102: Is MLC a machine learning method?

Author’s response: Indeed, MLC is a machine learning algorithm inside the ArcGIS environment. However, we changed the manuscript to avoid misunderstandings.

- What do you mean with “user-friendly machine learning technique? (Line 142).

Author’s response: Thank you for pointing this out. Indeed, the term “user-friendly” may lead to misunderstandings so it has been removed from the manuscript.

- What is 5 Ma? (Line 154)

Author’s response: Dear reviewer, Ma is an abbreviation for one million years ago.

- What’s the “following manuscript” as you state in Line 197?

Author’s response: We changed the word “manuscript” to “section”.

- What’s ranging from 2014 to 2016? (Line 220)

Author’s response: The ESM is a European scale product that is produced using images from the time period 2014 to 2016.

- Are the products generated using machine learning? (Line 221) Or the satellite images are classified with machine learning?

Author’s response: Thank you for pointing this out. The derived products have been classified satellite images using a machine learning algorithm (RF). We included the word “classified” inside the manuscript.

- SRTM or SRTM DEM? (Line 200)

Author’s response: Thank you for pointing this out. Correctly, it is a SRTM DEM.

- White cube houses (Line 257) or Greek-style white architecture?

Author’s response: Indeed, we are referring to Cycladic Greek-style white houses. We changed the text on the manuscript.

- The implementation of the RF algorithm was implemented?? (Line 280)

Author’s response: Thank you for highlight this out. We rephrase the text.

- You do your analysis in a GIS environment, but why particularly in ArcGIS? (Line 273)

Author’s response: Indeed, the most part of our processing and analysis was implemented on ArcGIS software. We refer the software’s name in order to clarify to the readers that we have used geo-processing tools that exist in this specified software. We are using also other GIS software such as QGIS, MapInfo but we are more experienced and familiar with ESRI’s products.

- Random Forest? Or is it Random Tree Forest (Line 411)?

Author’s response: Thank you for pointing this out. We removed the “Tree” from line 411.

- Satellite images? Satellite photos? Satellite pictures? Which one?

Author’s response: All are Satellite images. We have modified the text of the manuscript accordingly.

Recommendations on Issue 1: It is not easy to present scientific research in a foreign language. Meticulous use of language will ensure that your work is better perceived in the international community. It takes time to become an expert on this subject. Therefore, please do not misunderstand me for the examples I gave above. I just tried to illustrate that your text is difficult to understand. Please get a proofreading service from a native English speaker.

Author’s response: Dear reviewer, thank you for your understanding and your instructions. We take into account your suggestions and we tried to improve our manuscript.

Issue 2: I am not happy with the framing. I caught the word "thesis" somewhere in the article. My guess is that this is an article derived from a thesis. However, there is a serious writing technique difference between the thesis and the article. A thesis addressed to general readers who might have no prior knowledge of the subject. However, the articles are of interest to researchers with a certain knowledge of the subject. Therefore, we should not share information and images that we will not use for discussion in the article. For instance, Fig. 2 and 3 are irrelevant. The content in Fig. 8 is already acknowledged by the readers of this journal. Fig. 9 can be merged with Fig.4. Even though you classify your images with RF, you show a RF classification example on a tabular data set in Fig. 10. (I guess, we all know it). Other framing problems can be listed as:

- I'm not sure about the use of the term “Geospatial Intelligence”. At the same time, the term “machine learning” has been used. Do we need to emphasize geospatial intelligence even though machine learning is a sub-branch of artificial intelligence? Or do we mean the same thing with the terms geospatial intelligence and machine learning?

Author’s response: Dear author, first of all, thank you for your comments. The word thesis was a mistake and has been modified. In addition, Geospatial Intelligence (GEOINT) is a wider definition that is based on the combination of geospatial and EO data in order to interpret and generate new information/knowledge. Machine Learning as a type of AI methodology sometimes can be part also of GEOINT but that is not a rule and both could be applied separately. A future work might be the implementation of deep learning techniques as a complex subset of machine learning.

- Why did you need to describe the topographic features of Milos and Santorini in long sentences?

Author’s response: Dear reviewer, thank you for your comment, topographic features need to be described due to the general description/setting of the two islands in terms of better understanding regarding their natural environment.

- Subsection 2.3.2 is written like a thesis. Very long descriptions have been made of the software and data used. These parts need to be shortened.

Author’s response: Thank you for your comment, we have shortened some parts.

- In the flow of the text, you usually prefer to tell us what you did first, then how you did it. I think it should be the opposite.

Issue 3: There are some problems with the methodology presentation. - I understand from the text that you have downloaded two Sentinel-2 images from 2016 and 2021. You claim that you don't filter out the pixels that are clouds because of the white-colored architecture and the presence of open-pit mines. Since what you're looking at is urban areas, we can't expect a big increase in a year. The reflection of urban structures does not change like the vegetation, depending on the effects of the seasons. To solve the cloud problem, you could use "composite" images from those years using Google Earth Engine. You should explain why you chose to use a single image. Which images did you use? What are the dates?

Author’s response: Dear reviewer, thank you for your suggestions. We aware of Google Earth Engine but we are not familiar yet with the environment of the platform. We will consider your suggestions about the composite images because it is a very interesting idea. Indeed, we don’t filter out the images due to the reasons we described in our manuscript. The cloud masking is an extra step to eliminated potential locations that may covered from small clouds areas that possibly missed out due to the fact that are very thinly and may lead to misclassifications. As a result of this study, we found out that cloud masking is not feasible on Milos and Thira due to the similar reflection behavior between clouds and buildings. The images that we processed for our investigation were from ESA’s Sentinel - 2 mission (Copernicus Open Access Hub) with the following dates: For Thira (Santorini complex) 5/9/2016 & 29/9/2021and for Milos 18/9/2016 & 27/9/2021, respectively.

- The NDWI formula should be given with band names instead of band numbers. Band numbers vary by satellite platform. Like (Green-NIR)/(Green+NIR).

Author’s response: We have changed the formula following your instructions.

- Why do you present a monochromatic image in Fig. 6 (left)?

Author’s response: The left image on Fig. 6 is the NDWI product generated by SNAP software using grey scale colour bar. This colour range was selected in an effort to better highlight the distinction between land and water parts.

- Starting from 435, you state that the number of trees should not exceed 100, but you found the best accuracy with 300 trees. There are other sorts of hyperparameters to be tuned in the RF. We cannot generalize values for these hyperparameters since they may vary depending on the size, complexity, and collinearity of the data. Therefore, please mention “Hyperparameters Tuning” referencing some examples in the geomatics domain (Example 1, Example 2, Example 3), rather than mentioning “an iterative approach”.

Author’s response: Thank you for pointing this out. We modified the paragraph according to your comments.

- Can you elaborate on the R2 scores in Fig. 12?

Author’s response: Thank you for pointing this out. We have integrated and analysed this stat in the body of the manuscript.

- The accuracy assessment sub-section (3.2) contains the metrics that we should present in the methodology. In the result section, we shouldn’t give definitions.

Author’s response: Thank you for pointing this out. We modified the paragraph according to your comments. In particular we have create a new sub-section 2.3.4 titled as Classification Accuracy Validation.

- You can merge Table 3-4-5-6-7-8-9 into a single Table, just like the example below:

Author’s response: Dear reviewer, we followed your instructions regarding the merging of the aforementioned tables. In the updated version of the manuscript, we have created two new tables, each for every year (2016 & 2021) summarizing all the needed information from tables 3 to 9.

- In Fig. 15 caption, please write down what are (a), (b), (c), and (d).

Author’s response: Revised accordingly.

Round 2

Reviewer 4 Report

I carefully examined the revised version of the manuscript. The manuscript has been sufficiently improved according to my previous report. I think the manuscript is ready to go to print. I congratulate the authors for their substantial overhaul.

This manuscript is a resubmission of an earlier submission. The following is a list of the peer review reports and author responses from that submission.

Round 1

Reviewer 1 Report

The article addresses current challenges in the field of using existing technologies to record urbanization in coastal areas. It contains credible results and has all the necessary structural elements. However, I suggest the following additions:

- Lines 16 - 47: The abstract is very long and contains some general sentences that are not relevant. It also significantly exceeds the scope prescribed for the journal (200 words max). It needs to be shortened.

- Lines 50 - 132: The introductory chapter states the framework of the topics covered in the study. Since the central theme of the study is "Urbanization and Coastal Space", I suggest that you present a somewhat broader view of the spatial planning field here (rather than just being so explicit about satellite monitoring of land use impacts). Last but not least, the ICZM protocol itself (which you mention) is precisely for coordinated land use planning at the sea-land interface. Much has been written about this. It should also be taken into account (and mentioned) that the EU has co-funded a number of maritime and coastal planning projects over the last decades through various projects as well as projects to develop monitoring tools in coastal areas.

- Lines 230 - 394 (Methodology): It is clear from the above what the study is about (the methods and technologies used are described in detail), but I miss a short concise note along the lines of: what exactly is the problem you are addressing and what are the individual research questions? I suggest you add a few concise sentences to summarize.

- Line 231 (Methodology): Please explain in more detail how you followed the ICZM protocol, or exactly which of its provisions were followed and how?

- Line 238: Figure 4 - source?

- Lines 395 - 525 (Results): Did the analysis of urbanization expansion identify any deviations from existing land use plans? If this was not the subject of the study, I suggest you write this.

- I suggest that you elaborate in the Discussion (or Conclusions) chapter on how these technologies can be useful for the spatial planning process itself, not just the process of monitoring the situation in space.

Reviewer 2 Report

This manuscript delineated coastal zone and related settlements in two Greece islands. The main contribution of this manuscript is to provide a framework for defining coastal zones using earth observation data. The paper is overall well written and interesting, but there are some flaws in the manuscript that need to be addressed before publication:

(1)   Although the title is “Geospatial Intelligence and Machine Learning Technique for Urban Mapping in Coastal Regions of South Aegean Volcanic Arc Islands”, we don’t find enough information on Geospatial Intelligence in this paper. The authors just used a common supervised classification in mapping coastal settlements . In this point of view, the title is somewhat misleading.

(2)   Abstract: Lacking a brief description of current knowledge gaps in the related fields.

(3)   Line 215: The authors only mentions the GHSL method but not provide any details about it.

(4)   Line 284: A 0-threshold of NDWI method was used in separating water and non-water. Since classification performances are highly dependent on the used water index, the authors should provide experiments on selecting the water index .

(5)   Line 308: While the authors followed the existing methodology, it should be explained more about how did they retrieve the bathymetry as this is a main part of this study.

(6)   Lines 363-365: A scatter-plot figure should be given here to support the poor correlation between derived water depth and the charted depths above the 10 m.

(7)   Line 367: Confused abbreviations. “Random-tree Forest” has been abbreviated to “RT” in the line 107, but in this line it was abbreviated to “RF”. It is recommended to unify the abbreviations of Random-tree Forest in the full text (RT or RF).

(8)   It is recommended to combine Section 3.2 and Section 3.4 as they are both about result validation.

(9)   The authors should provide the number of training samples, sample collection methods, the number of sample classes, and which input features for random forest classification.

(10) Figs 12-16: These figures could be merged because they have the same topic.

(11) The discussion part is a bit weak and confusing. It is suggested to discuss the key findings, practical significance, and limitations & next steps in this section.

(12) The full names of some abbreviations are not given, such as “GEBCO” (line 32) and “GHSL” (line 215).

Reviewer 3 Report

Please check the author requirements for this journal: The abstract is too long (recommended length: 200 words) and the aims and results of the study are not clear enough.

In the Introduction, the authors go into a lot of detail about the geomorphology of the area without making the connection to urban areas and why this is important to this study. A description of the actual urban areas in the study area is missing in the Introduction. Furthermore, the introduction is missing some sources. For example, where do the figures from ll. 62-64 come from?

The Materials and Methods section does not provide all necessary information. For example, the subtraction of the road network from the products using OSM data is not mentioned.

A large part of the Discussion repeats information from the Introduction, rather than discussing the findings of the study. It would be interesting to include a population estimate to further quantify how the respective areas are affected.

Also, the citations are not consistent (esp. in the Materials and Methods chapter).

The language (spelling, punctuation and grammar) needs to be double checked, there are a number of typographical errors in the manuscript (“alos” instead of “also”, “Sout” instead of “South", “are” instead of “area” etc.) and the wording is not always correct. ALL chapters need to be extensively revised (check for issues like “free software… is provided free of charge” and introductory phrases like “on the contrary”, “specifically” etc.) and proof-read.

The authors often use a comma separator for percentages and other values instead of a point separator.

Overall, a connection between the three methods is not obvious. Why is the bathymetry necessary for urban mapping? To classify the expansion of urban areas, this does not seem to add any value.

General comments:

Line 19: “operations” is not the right word.

Line 32: all abbreviations should be spelt in full (GEBCO).

Lines 60-62: this sentence needs clarification. Being critical AND important sounds like a redundancy.

Lines 107: RT or RF? 

Lines 168ff: Santorini and Thira are used interchangeably which is confusing to read.

Line 191: “as long as” does not make sense here.

Line 206: “thus” is not the right word here.

Line 226-228: this is confusing - primary and main road network? 

Line 261: not clear at this stage why this was repeated 40 times. A reference to chap. 2.3.3. would be useful.

Line 276: what does “regarding Sentinel 2 images” mean here? Please check the use of “regarding” throughout the paper.

Line 291: what is meant by “correspondence literature”? This is not the right word here.

Line 295: this is not entirely clear.

Line 296: “on the contrary” is not the right phrase here.

Line 317: IKONOS was retired in 2015 so no imagery from 2016 or 2021 can be obtained.

Line 346: was this model developed for this manuscript or for another thesis?

Lines 356-357: why does it become unreliable? And how is the extinction depth critical to the reliability? Explain further or provide sources.

Lines 391-394: more detail on how this assessment was performed.

Lines 392-394: why 70% and 30%?

Lines 447-452: evidence of this (in form of a map showing examples) would help here.

Line 507: check position of “<“

Lines 574-580: if cloud-free images were used in the first place, how can this be a finding of this study? There is no mention of using cloud masks in the method.

Lines 582-587: this fits better to the previous chapter.

Figures:

Figures 1-3: please insert North arrows.

Figures 2 and 3: the image credits in the grey boxes at the bottom are illegible.

Figure 4: it is not clear from this image which way the arrows from the ESM dataset and Classified Raster goes.

Figure 9: the SDB values are too small to read.

Figure 17: needs further description in the caption. The figure is very hard to interpret, esp. 17c and 17f are not clear.

Tables: 

Tables 2 and 3: change description so it is obvious which island is referred to.